

**An intercomparison of oceanic methane and nitrous oxide measurements**
Samuel T. Wilson[1*], Hermann W. Bange[2], Damian L. Arévalo-Martínez[2], Jonathan Barnes[3],
Alberto V. Borges[4], Ian Brown[5], John L. Bullister[6], Macarena Burgos[7], David W. Capelle[8],
Michael Casso[9], Mercedes de la Paz[10†], Laura Farías[11], Lindsay Fenwick[8], Sara Ferrón[1], Gerardo
Garcia[11], Michael Glockzin[12], David M. Karl[1], Annette Kock[2], Sarah Laperriere[13], Cliff S.
Law[14,15], Cara C. Manning[8], Andrew Marriner[14], Jukka-Pekka Myllykangas[16], John W.
Pohlman[9], Andrew P. Rees[5], Alyson E. Santoro[13], Mabel Torres[11], Philippe D. Tortell[8], Robert
C. Upstill-Goddard[3], David P. Wisegarver[6], Guiling L. Zhang[17], Gregor Rehder[12]
[1]University of Hawai'i, Daniel K. Inouye Center for Microbial Oceanography: Research and
Education (C-MORE), Honolulu, Hawai'i, USA
[2] GEOMAR Helmholtz Centre for Ocean Research Kiel, Düsternbrooker Weg 20 24105 Kiel,
Germany
[3]Newcastle University, School of Natural and Environmental Sciences, Newcastle upon Tyne,
UK
[4]Université de Liège, Unité d'Océanographie Chimique, Liège, Belgium
[5]Plymouth Marine Laboratory, Plymouth, UK
[6]National Oceanic and Atmospheric Administration, Pacific Marine Environmental Laboratory,
Seattle, Washington, USA
[7] Universidad de Cádiz, Instituto de Investigaciones Marinas, Departmento Química-Física
Cádiz, Spain
[8]University of British Columbia, Vancouver, Department of Earth, Ocean and Atmospheric
Sciences, British Columbia, Canada
[9]US Geological Survey, Woods Hole Coastal and Marine Science Center, Woods Hole, USA
[10]Instituto de Investigaciones Marinas, Vigo, Spain
[11]University of Concepción, Department of Oceanography, Laboratory of Oceanographic
Process and Climate (PROFC), Concepción, Chile
[12]Leibniz Institute for Baltic Sea Research Warnemünde, Rostock, Germany
[13]University of California Santa Barbara, Department of Ecology, Evolution, and Marine
Biology, Santa Barbara, USA
[14]National Institute of Water and Atmospheric Research (NIWA), Wellington, New Zealand

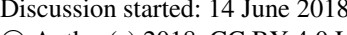



[15]Department of Chemistry, University of Otago, Dunedin, New Zealand
[16]University of Helsinki, Department of Environmental Sciences, Helsinki, Finland
[17]Ocean University of China, Department of Marine Chemistry, Qingdao, China
[†]Current address: Instituto Español de Oceanografía, Centro Oceanográfico de A Coruña, A
Coruña, Spain

*corresponding author: stwilson@hawaii.edu



**Abstract.** Large scale climatic forcing is impacting oceanic biogeochemical cycles and is
expected to influence the water-column distribution of trace gases including methane and nitrous
oxide. Our ability as a scientific community to evaluate changes in the water-column inventories
of methane and nitrous oxide depends largely on our capacity to obtain robust and accurate
concentration measurements which can be validated across different laboratory groups. This
study represents the first formal, international, intercomparison of oceanic methane and nitrous
oxide measurements whereby participating laboratories received batches of seawater samples
from the subtropical Pacific and the Baltic Sea. Additionally, compressed gas standards from the
same calibration scale were distributed to the majority of participating laboratories to improve
the analytical accuracy of the gas measurements. The computations used by each laboratory to
derive the dissolved gas concentrations were also evaluated for inconsistencies (*e.g.* pressure and
temperature corrections, solubility constants). The results from the intercomparison and
intercalibration exercises provided invaluable insights into methane and nitrous oxide
measurements. It was observed that analyses of seawater samples with the lowest concentrations
of methane and nitrous oxide had the lowest precisions. In comparison, while the analytical
precision for samples with the highest concentrations of trace gases was better, the variability
between the different laboratories was higher; 36% for methane and 27% for nitrous oxide. In
addition, the comparison of different batches of seawater samples with methane and nitrous
oxide concentrations that ranged over an order of magnitude revealed the ramifications of
different calibration procedures for each trace gas. Overall, this paper builds upon the
intercomparison results to develop a framework for improving oceanic methane and nitrous
oxide measurements, with the aim of precluding future analytical discrepancies between
laboratories.



## 1. Introduction

The increasing mole fractions of greenhouse gases in the Earth's atmosphere are causing long-term climate change with unknown future consequences. Two greenhouse gases, methane and nitrous oxide, together contribute approximately 23% of total radiative forcing attributed to well-mixed greenhouse gases (Myhre et al., 2013). It is imperative that the monitoring of methane and nitrous oxide in the Earth's atmosphere is accompanied by measurements at the Earth's surface to better inform the sources and sinks of these climatically important trace gases. This includes measurements of dissolved methane and nitrous oxide in the marine environment, which is an overall source of both gases to the overlying atmosphere (Nevison et al., 1995; Anderson et al., 2010; Naqvi et al., 2010; Freing et al., 2012; Ciais et al., 2014).

Oceanic measurements of methane and nitrous oxide are conducted as part of established time-series locations, along hydrographic survey lines, and during disparate oceanographic expeditions. Within low to mid-latitude regions of the open ocean, the surface waters are typically slightly super-saturated with respect to atmospheric equilibrium for both methane and nitrous oxide. There is typically an order of magnitude range in concentration along a vertical water-column profile at any particular open ocean location (e.g. Wilson et al., 2017). In contrast to the open ocean, near-shore environments, which are subject to river inputs, coastal upwelling, benthic exchange and other processes, have higher concentrations and greater spatial and temporal heterogeneity (e.g. Schmale et al., 2010; Upstill-Goddard and Barnes, 2016).

Methods for quantifying dissolved methane and nitrous oxide have evolved and somewhat diverged since the first measurements were made in the 1960s (Craig and Gordon 1963; Atkinson and Richards 1967). Some laboratories employ purge-and-trap methods for extracting and concentrating the gases prior to their analysis (*e.g.* Zhang et al., 2004; Bullister and Wisegarver, 2008; Capelle et al., 2015; Wilson et al., 2017). Others equilibrate a seawater sample with an overlying headspace gas and inject a fixed volume of the gaseous phase into a gas analyzer (e.g. Upstill-Goddard et al., 1996; Walter et al., 2005; Farias et al., 2009). Additional developments for continuous underway surface seawater measurements use equilibrator systems of various designs coupled to a variety of detectors (*e.g.* Weiss et al., 1992; Butler et al., 1989; Gülzow et al., 2011; Arévalo-Martínez et al., 2013). Determining the level of analytical comparability between different laboratories for discrete samples of methane and nitrous oxide is an important step towards improved comprehensive global assessments. Such



intercomparison exercises are critical to determining the spatial and temporal variability of
methane and nitrous oxide across the world oceans with confidence, since no single laboratory
can single-handedly provide all the required measurements at sufficient resolution. Previous
comparative exercises have been conducted for other trace gases *e.g.* carbon dioxide,
dimethylsulphide, and sulfur hexafluoride (Dickson et al., 2007; Bullister and Tanhua, 2010;
Swan et al., 2014) and for trace elements (Cutter et al., 2013). These exercises confirm the value
of the intercomparison concept.

To instigate this process for methane and nitrous oxide, a series of international

intercomparison exercises were conducted between 2013 and 2017, under the auspices of
Working Group #143 of the Scientific Committee on Oceanic Research (SCOR) (www.scor-
int.org). Discrete seawater samples collected from the subtropical Pacific Ocean and the Baltic
Sea were distributed to the participating laboratories (Table 1). The samples were selected to
cover a representative range of concentrations across marine locations, from the oligotrophic
open ocean to highly productive waters, and in some instances sub-oxic, coastal waters. An
integral component of the intercomparison exercise was the production and distribution of
methane and nitrous oxide gas standards to members of the SCOR Working Group. The
intercomparison exercise was conceived and evaluated with the following four questions in
mind:
Q1. What is the agreement between the SCOR gas standards and the 'in-house' gas standards

used by each laboratory?

Q2. How do measured values of dissolved methane and nitrous oxide compare across

laboratories?

Q3. Despite the use of different analytical systems, are there general recommendations to reduce

uncertainty in the accuracy and precision of methane and nitrous oxide measurements?

Q4. What are the implications of inter-laboratory differences for determining the spatial and

temporal variability of methane and nitrous oxide in the oceans?


**2. Methods**
**2.1 Calibration of nitrous oxide and methane using compressed gas standards**
Laboratory-based measurements of oceanic methane and nitrous oxide require separation of the
dissolved gas from the aqueous phase, with the analysis conducted on the gaseous phase.



Calibration of the analytical instrumentation used to quantify the concentration of methane and
nitrous oxide is nearly always conducted using compressed gas standards, the specifics of which
vary between each laboratory.  Therefore, the reporting of methane and nitrous oxide datasets
ought to be accompanied by a description of the standards used, including their methane and
nitrous oxide mole fractions, the declared accuracies, and the composition of their balance or
'make-up' gas.  For both gases, the highest accuracy commercially available standards have
mole fractions close to current day atmospheric values.  These standards can be obtained from
national agencies e.g. National Oceanic and Atmospheric Administration Global Monitoring
Division (NOAA GMD), the National Institute of Metrology China, and the Central Analytical
Laboratories of the European Integrated Carbon Observation System Research Infrastructure
(ICOS-RI).  By comparison, it is more difficult to obtain highly accurate methane and nitrous
oxide gas standards with mole fractions exceeding modern-day atmospheric values.  This is
particularly problematic for nitrous oxide due to the nonlinearity of the widely used Electron
Capture Detector (ECD) (Butler and Elkins, 1991).
The absence of a widely available high mole fraction, high accuracy nitrous oxide gas
standard was noted as a primary concern at the outset of the intercomparison exercise.
Therefore, a set of high-pressure primary gas standards was prepared for the SCOR Working
Group by John Bullister and David Wisegarver at NOAA Pacific Marine and Environmental
Laboratory (PMEL).  One batch, referred to as Air Ratio Standard (ARS), had methane and
nitrous oxide mole fractions similar to modern air and the other batch, referred to as Water Ratio
Standard (WRS) had higher methane and nitrous oxide mole fractions for calibration of high
concentration water samples.  These SCOR primary standards were checked for stability over a
12 month period and assigned mole fractions on the same calibration scale, known as 'SCOR-
2016.'  A comparison was conducted with NOAA standards prepared on the SIO98 calibration
scale for nitrous oxide and the NOAA04 calibration scale for methane.  Based on the comparison
with NOAA standards, the uncertainty of the methane and nitrous oxide mole fractions in the
ARS and the uncertainty of the methane mole fraction in the WRS were all estimated at better
than 1%.  By contrast, the uncertainty of the nitrous oxide mole fraction in the WRS was
estimated at 2-3%.  The gas standards were distributed to twelve of the laboratories involved in
this study (Table 1).  A technical report on the production of the gas standards and their assigned
absolute mole fractions is available at www.scor-int.org/SCOR_Publications.






## 2.2 Collection of discrete samples of nitrous oxide and methane

Dissolved methane and nitrous oxide samples for the intercomparison exercise were collected
from the subtropical Pacific Ocean and the Baltic Sea. Pacific samples were obtained on 28
November 2013 and 24 February 2017 from the Hawaii Ocean Time-series (HOT) long-term
monitoring site, Station ALOHA, located at 22.75 N, 158.00 W. The November 2013 samples
are included in Figure S1 and S2 in the Supplement, but are not discussed in the main Results or
Discussion because fewer laboratories were involved in the initial intercomparison, and the
results from these samples support the same conclusions obtained with the more recent sample
collections. Seawater was collected using Niskin-like bottles designed by John Bullister (NOAA
PMEL), which help minimize contamination of trace gases, in particular chlorofluorocarbons
and sulfur hexafluoride (Bullister and Wisegarver, 2008). The bottles were attached to a rosette
with a conductivity-temperature-depth (CTD) package. Seawater was collected from two depths:
700 m and 25 m, where the near-maximum and minimum water-column concentrations for
methane and nitrous oxide at this location can be found. Replicate samples were collected from
each bottle, with one replicate reserved for analysis at the University of Hawaii to evaluate
variability between sampling bottles. Seawater was dispensed from the Niskin-like bottles using
Tygon® tubing into the bottom of borosilicate glass bottles, allowing overflow of at least two
sample volumes and ensuring the absence of bubbles. Most sample bottles were 240 mL in size
and were sealed with no headspace using butyl-rubber stoppers and aluminum crimp-seals. A
few laboratory groups requested smaller crimp-sealed glass bottles ranging from 20-120 mL in
volume and two laboratories used 1 L glass bottles which were closed with a stopper and sealed
with Apiezon® grease. Seawater samples were collected in quadruplicate for each laboratory.
All samples were preserved using saturated mercuric chloride solution (100 μL of saturated
mercuric chloride solution per 100 mL of seawater sample) and stored in the dark at room
temperature until shipment.
Samples from the western Baltic Sea were collected during 15-21 October 2016, onboard the
R/V *Elisabeth Mann Borgese* (Table 2). Since the Baltic Sea consists of different basins with
varying concentrations of oxygen beneath permanent haloclines (Schmale et al., 2010), a larger
range of water-column methane and nitrous oxide concentrations were accessible for inter-
laboratory comparison compared to Station ALOHA. For all seven Baltic Sea stations, the



water-column was sampled into an on-deck 1,000 L water tank that was subsequently
subsampled into discrete sample bottles.  At three stations (BAL1, BAL3, and BAL6), the water
tank was filled from the shipboard high-throughput underway seawater system.  For deeper
water-column sampling at the stations BAL2, BAL4, and BAL5, the water tank was filled using
a pumping CTD system (Strady et al., 2008) with a flow rate of 6 L min$^{-1}$ and a total pumping
time of approximately 3 h.  For the final deep water-column station, BAL7, the pump that
supplied the shipboard underway system was lowered to a depth of 21 m to facilitate a shorter
pumping time of approximately 20 mins.  Subsampling the water tank for all samples took
approximately 1 h in total and the total sampling volume was less than 100 L.  To verify the
homogeneity of the seawater during the sampling process, the first and last samples collected
from the water tank were analyzed by Newcastle University onboard the research vessel.  In
contrast to the Pacific Ocean sampling, which predominantly used 240 mL glass vials, each
laboratory provided their own preferred vials and stoppers for the Baltic Sea samples.  Seawater
samples were collected in triplicate for each laboratory.  All samples were preserved with 100
μL of saturated mercuric chloride solution per 100 ml of seawater sample, with the exception of
samples collected by U.S. Geological Survey, who analyzed unpreserved samples onboard the
research vessel.

**2.3. Sample analysis**
Each laboratory measured dissolved methane and nitrous oxide slightly differently.  A full
description of each laboratory's method is not included in the main document and instead can be
found in Table S6 and Table S7 in the Supplement for methane and nitrous oxide, respectively.

The majority of laboratories measured methane and nitrous oxide by equilibrating the

seawater sample with an overlying headspace and subsequently injecting a portion of the gaseous
phase into the gas analyzer.  This method has been conducted since the 1960s when gas
chromatography was first used to quantify dissolved hydrocarbons (McAuliffe, 1963).  The
headspace was created using helium, nitrogen, or high-purity air to displace a portion of the
seawater sample within the sample bottle.  Alternatively, a subsample of the seawater was
transferred to a gas-tight syringe and the headspace gas subsequently added.  The volume of the
vessel used to conduct the headspace equilibration ranged from 20 ml borosilicate glass vials to 1
L glass vials and syringes used by Newcastle University and U.S. Geological Survey,





respectively.  The dissolved gases equilibrated with the overlying headspace at a controlled
temperature for a set period of time that ranged from 20 min to 24 h.  The equilibration process
was typically enhanced by some initial period of physical agitation.  After equilibration, an
aliquot of the headspace was transferred into the gas analyzer (GA) by either physical injection,
displacement using a brine solution, or injection using a switching valve.  Alternatively, a
subsample of the headspace was collected into a gas tight syringe and subsequently injected into
the GA.  Some laboratories incorporated a drying agent and a carbon dioxide scrubber prior to
analysis.  The gas sample passed through a multi-port injection valve containing a sample loop of
known volume, which transferred the gas sample directly onto the analytical column within the
oven of the GA.  Calibration of the instrument was achieved by passing the gas standards
through the injection valve.
The final gas concentrations using the headspace equilibration method was calculated by:

[1]  $$C_{gas} \text{ [nmol L}^{-1}] = \left( \beta \; x \; PV_{wp} + \frac{xP}{RT}V_{hs} \right)/V_{wp}$$

where $\beta$ is the Bunsen solubility of nitrous oxide (Weiss and Price, 1980) or methane
(Wiesenburg and Guinasso, 1979) in nmol L$^{-1}$ atm$^{-1}$, $x$ is the dry gas mole fraction (ppb)
measured in the headspace, $P$ is the atmospheric pressure (atm), $V_{wp}$ is the volume of water
sample (mL), $V_{hs}$ is the volume (mL) of the created headspace, $R$ is the gas constant (0.08205746
L atm K$^{-1}$mol$^{-1}$), and $T$ is equilibration temperature in Kelvin (K).  An example calculation is
provided in Table S8 in the Supplement.
In contrast to the headspace equilibrium method, five laboratories used a purge-and-trap
system for methane and/or nitrous oxide analysis (Table S6 and Table S7 in the Supplement).
These systems were directly coupled to a Flame Ionization Detector (FID) or ECD, with the
exception of University of British Columbia, where a quadrupole mass spectrometer with an
electron impact ion source and Faraday cup detector were used (Capelle et al., 2015).  The
purge-and-trap systems were broadly similar, each transferring the seawater sample to a sparging
chamber.  Sparging times typically ranged from 5-10 min and the sparge gas was either high
purity helium or high purity nitrogen.  Further purification of the sparge gas was achieved prior
to use by passing it through tubing packed with Poropak Q and maintained at low temperatures.
This is a requirement to achieve a low blank signal.  The elutant gas was dried using Nafion or



Drierite, and subsequently cryotrapped on a sample loop packed with Porapak Q to aid retention
of methane and nitrous oxide. Cryotrapping was achieved using liquid nitrogen (-165$^{o}$C) or
cooled ethanol (-70$^{o}$C). Subsequently, the valve was switched to inject mode and the sample
loop was rapidly heated to transfer its contents onto the analytical column. Calibration was
achieved by injecting standards via sample loops using multi-port injection valves. Injection of
standards prior to the sparging chamber allowed for calibration of the purge-and-trap gas
handling system, in addition to the GA. Calculation of the gas concentrations using the purge-
and-trap method was achieved by application of the ideal gas law to the standard gas
measurements:
[2]        $PV = nRT$
where $P$, $R$, and $T$ are the same as Equation 1, $V$ represents the volume of gas injected (L),
and $n$ represents moles of gas injected. Rearranging Equation 2 yields the number of moles of
methane or nitrous oxide gas for each sample loop injection of compressed gas standards. These
values were used to derive a calibration curve based on the measured peak areas of the injected
standards, in order to derive the number of moles measured for each unknown sample. To
calculate concentrations of methane or nitrous oxide in a water sample, the number of moles
measured were divided by the volume (L) of seawater sample analyzed. An example calculation
is provided in Table S8 in the Supplement.

**2.4 Data analysis**

The final concentrations of methane and nitrous oxide are reported in nmol kg$^{-1}$. The analytical
precision for each batch of samples obtained by each of the individual laboratories was estimated
from the analysis of replicate seawater samples and reported as the coefficient of variation (%).
The values reported by each laboratory for all the batches of seawater samples are shown in
Tables S1 to S4 in the Supplement. Due to the observed inter-laboratory variability, it is likely
that the median value of methane and nitrous oxide for each batch of samples does not represent
the absolute *in situ* concentration. As this complicates the analytical accuracy for each
laboratory, we instead calculated the percentage difference between the median concentration
determined for each set of samples and the mean value reported by an individual laboratory. The
presence of outliers was established using the Interquartile Range (IQR) and by comparing with
one standard deviation applied to the overall median value.




## 3. Results

### 3.1 Comparison of methane and nitrous oxide gas standards

Six laboratories compared their existing 'in-house' standards of methane with the SCOR

standards. This was done by calibrating in-house standards and deriving a mixing ratio for the

SCOR standards which were treated as unknowns. Four laboratories reported methane values for

either the ARS or WRS within 3% of their absolute concentration, whereas two laboratories

reported an offset of 6% and 10% between their in-house standards and the SCOR standards

(Table S6 in the Supplement). For those laboratories who measured the SCOR standards to

within 3% or better accuracy, observed offsets in methane concentrations from the overall

median cannot be due to the calibration gas.

Seven laboratories compared their own in-house standards of nitrous oxide with the prepared

SCOR standards. Six laboratories reported values of nitrous oxide for the ARS which were

within 3% of the absolute concentration, with the remaining laboratory reporting an offset of

10% (Table S7 in the Supplement). The majority of these laboratories (five out of six groups)

compared the SCOR ARS with NOAA GMD standards, which have a balance gas of air instead

of nitrogen. Some laboratories with analytical systems that incorporated fixed sample loops (*e.g.*

1 or 2 ml loops housed in a 6-port or 10-port injection valve) had difficulty analyzing the WRS,

as the peak areas created by the high mole fraction of the standard exceeded the signal typically

measured from in-house standards or acquired by sample analysis, by an order of magnitude.

The high mole fraction of the WRS was not an issue when multiple sample loops of varying

sizes were incorporated into the analytical system, which was the case for purge-and-trap based

designs. For the two laboratories which had comparable values of their in-house standard and

the WRS, an offset of 3% and a >20% offset was reported.

### 3.2 Methane concentrations in the intercomparison samples

Overall, median methane concentrations in seawater samples collected from the Pacific

Ocean and the Baltic Sea ranged from 0.9 to 60.3 nmol kg$^{-1}$ (Table 2). Out of 101 reported

values, 3 outliers were identified using the IQR criterion and were not included in further

analysis. The methane data values for each batch of samples analyzed by each laboratory,



including the mean and standard deviation, the number of samples analyzed, and the % offset
from the overall median value are reported in Table S1 and Table S2 in the Supplement.
The two Pacific Ocean sampling sites had the lowest water-column concentrations of
methane (Fig. 1a and 1b).  The PAC1 samples collected from within the mesopelagic zone,
where methane concentrations have been reported to be less than 1 nmol kg$^{-1}$ (Reeburgh et al.,
2007; Wilson et al., 2017), showed a distribution of reported concentrations skewed towards the
higher values.  For the PAC1 samples, 7 out of 12 laboratories reported values ≤1 nmol kg$^{-1}$ and
the mean coefficient of variation for all laboratories was 11% (Table 2).  In contrast to the
mesopelagic samples, the methane concentrations for the near-surface seawater samples (PAC2)
were close to atmospheric equilibrium (Fig. 1b).  Measured concentrations of methane for PAC2
samples ranged from 1.9 to 3.8 nmol kg$^{-1}$ and the mean coefficient of variation for all
laboratories was 7%.  Similar to the PAC1 samples, PAC2 also had a distribution of data skewed
towards the higher concentrations.
Three Baltic Sea sampling sites (BAL1, BAL3, and BAL6) had median methane
concentrations that ranged from 4.1 to 5.7 nmol kg$^{-1}$ (Fig. 1c).  The BAL1 samples also showed a
skewed distribution of reported values towards higher concentrations, as seen in PAC1 and
PAC2 samples. However, this was not evident in BAL3 or BAL6, which have a higher
agreement between the reported methane concentrations.  For these three sets of Baltic Sea
samples, the mean coefficient of variation for all laboratories ranged from 4% (BAL3) to 9%
(BAL1).  The next three Baltic Sea samples (BAL4, BAL5, and BAL7) had methane
concentrations that ranged from 18.8 to 35.4 nmol kg$^{-1}$ (Fig. 1d).  These three sets of samples
had a normal distribution of data and the highest agreement between the reported concentrations
for all of the Pacific Ocean and Baltic Sea samples.  Furthermore, for these three sets of samples,
the mean coefficient of variation for all laboratories was 4% (Table 2).  The final Baltic Sea
sample (BAL2) had the highest concentrations of methane, with a median reported value of 60.3
nmol kg$^{-1}$, and a large range of values (45.2 to 67.2 nmol kg$^{-1}$; Fig. 1e).  The BAL2 samples had
the lowest overall mean coefficient of variation for all laboratories; 2% (Table 2).
Further analysis of the data was conducted to better comprehend the factors that caused the
observed inter-laboratory variability in methane measurements.  The deviation from median
values was calculated for each sample collected from the Baltic Sea (Fig. 2).  The Pacific Ocean
samples (PAC1 and PAC2) were not included in this analysis due to the skewed distribution of



data.  There were also some instances in the Baltic Sea samples, where the median concentration
might not have realistically represented the absolute *in situ* methane concentration.  This was
most likely to have occurred at low concentrations due to the skewed distribution of reported
concentrations (*e.g.* BAL1) or at high concentrations where there was a large range in reported
values (*e.g.* BAL2).  The results revealed that a few laboratories (Datasets D, F, and G) were
consistently within or close to 5% of the median value for all batches of seawater samples (Fig.
2).  Some laboratories (e.g. Datasets B, C, and H) had a higher deviation from the median value
at higher methane concentrations.  Two laboratories (Datasets J and K) had a higher deviation
from the median value at lower methane concentrations.  Finally, in some cases it was not
possible to determine a trend (Datasets A and E), due to the variability.

The reasons behind the trends for each dataset became more apparent when considering the

response of the FID at nanomolar concentrations of methane and a 'typical' calibration curve
(Fig. 3).  The FID has a linear response to methane at nanomolar values and therefore a high
level of accuracy across a relatively wide range of *in situ* methane concentrations can be
obtained with the correct slope and intercept.  To demonstrate this, calibration curves for
methane were provided by the University of Hawaii.  These revealed minimal variation in the
slope value when calibration points were increased from low mole fractions (Fig. 3a) to higher
mole fractions (Fig. 3b).  However, the intercept value was sensitive to the range of calibration
values used, and this effect was further exacerbated when only the higher calibration points were
included (*i.e.* Fig. 3c).  The relevance to final methane concentrations is demonstrated by
considering the PAC2 samples reported by the University of Hawaii.  A measured peak area of
62 for a sample volume of 0.076 L and a seawater density of 1024 kg m$^{-3}$, yields final methane
concentrations of 2.1, 2.2, and 2.8 nmol kg$^{-1}$ depending on whether the equations from Fig.3a,
3b, or 3c are used, respectively.  Therefore, an almost 30% increase in final methane
concentration results from use of the equation in Figure 3c, compared to Figure 3a.  With this
understanding on the effect of FID calibration, we consider it likely that the increased deviation
from median values at high methane concentrations (Datasets B, C, and H) results from
differences in calibration slope between each laboratory.  In contrast, the datasets with a higher
offset at low methane concentrations could be due to the use of incorrect intercepts as well as
other factors including sample contamination, discussed below (Datasets J and K).



### 3.3 Nitrous oxide concentrations in the intercomparison samples


Overall, median nitrous oxide concentrations in seawater samples collected from the Pacific
Ocean and the Baltic Sea ranged from 3.4 to 42.4 nmol kg$^{-1}$ (Table 2). Of the 113 reported
values, ten outliers were identified using the IQR criterion and were not included in further
analysis. The nitrous oxide data values for each batch of samples analyzed by each laboratory,
including the mean and standard deviation, the number of samples analyzed, and the % offset
from the overall median value are reported in Table S3 and Table S4 in the Supplement.
For six sets of seawater samples, BAL1, BAL2, BAL3, BAL6, BAL7, and PAC2, the
concentrations of nitrous oxide were close to atmospheric equilibrium. The reported values
ranged from 7.7 to 12.7 nmol kg$^{-1}$ in the Baltic Sea (Fig. 4a) and from 5.9 to 7.6 nmol kg$^{-1}$ in the
Pacific Ocean (Fig. 4b). For the Pacific Ocean near-surface sampling site (PAC2), the
theoretical value of nitrous oxide concentration in equilibrium with the overlying atmosphere is
also shown (Fig. 4b). For these six samples with concentrations close to atmospheric
equilibrium, the mean coefficient of variation for all laboratories ranged from 3% (BAL3 and
PAC2) to 5% (BAL1) (Table 2).
For the three other sets of samples (BAL4, BAL5, and PAC1), the nitrous oxide
concentrations deviated significantly from atmospheric equilibrium (Fig. 4c, 4d, and 4e). At one
sampling site, BAL4 (Fig. 4c), nitrous oxide was under-saturated with respect to atmospheric
equilibrium and reported concentrations ranged from 2.1–5.5 nmol kg$^{-1}$. As observed in the low
concentration Pacific Ocean methane samples, there was a skewed distribution of the data
towards the higher nitrous oxide concentrations. The BAL4 samples also had the highest
variability (*i.e.* lowest precision), with a mean coefficient of variation of 8% (Table 2). The two
remaining samples (PAC1 and BAL5) had much higher concentrations of nitrous oxide, as
expected for low-oxygen regions of the water-column. In contrast to the samples with near
atmospheric equilibrium concentrations of nitrous oxide, there was a low overall agreement
between the independent laboratories for PAC1 and BAL5 nitrous oxide concentrations (Fig. 4d,
4e). At PAC1 and BAL5, reported nitrous oxide concentrations ranged from 34.3–45.8 nmol kg$^{-}$
$^{1}$ (Fig. 4d) and 30.1–45.9 nmol kg$^{-1}$, respectively (Fig. 4e). The mean coefficient of variation for
all laboratories was 4% for BAL5 samples compared to 3% for PAC1 samples.
The deviation from median value was analyzed for the nitrous oxide datasets to gain a deeper
insight into the variability associated with their measurements (Fig. 5). The BAL1 dataset was



not included in this analysis due to its skewed data distribution and the high inter-laboratory
variability for BAL5 indicated that the median value may differ from the absolute nitrous oxide
concentration for this sample.  For the low nitrous oxide Baltic Sea and Pacific Ocean samples
(Fig. 5a), the majority of data points were within 5% of the median values.  Furthermore, for the
majority of laboratories, the data points for separate seawater samples clustered together
indicating some consistency to the extent they varied from the overall median value.  Exceptions
to this observation include Datasets E, C, L, and K (Fig. 5a) which demonstrated varying
precision and accuracy.  At high nitrous oxide concentrations (Fig. 5b), there are fewer data
points within 5% of the median value compared to low nitrous oxide concentrations (Fig. 5a).
Therefore, for PAC1 and BAL5 samples, 6 and 7 data points fall within 5% of the median value,
respectively.  Furthermore, only three laboratories (Datasets F, G, and K) had data for both
Pacific Ocean and Baltic Sea samples within 5% of the median value.  This could have been
caused by inconsistent analysis between different batches of samples or by variable sample
collection and transportation.
The likely factors that caused these offsets in nitrous oxide concentrations among
laboratories include sample analysis and calibration of the gas analyzers.  Calibration of the ECD
is nontrivial and at least two prior publications have discussed nitrous oxide calibration issues
(Butler and Elkins, 1990; Bange et al., 2001).  The laboratories participating in the nitrous oxide
intercomparison employed different calibration procedures (Fig. 6).  Some used a linear fit and
maintained their analytical peak areas within a narrow range (Fig. 6a), while others used a step-
wise linear fit and therefore used different slopes for low and high nitrous oxide mole fractions
(Fig. 6b).  Finally, some applied a polynomial curve (Fig. 6c) and sometimes two different
polynomial fits, for low and high concentrations.  The difficulty in calibrating the ECD was
evidenced by the deviation from median values as multiple datasets show good precision but
consistent offsets at the lowest (Fig. 5a) and highest (Fig. 5b) final concentrations of nitrous
oxide.
**3.4 Sample storage**
Because prolonged sample storage adversely affects dissolved methane and nitrous oxide
samples (Magen et al., 2014), the intercomparison datasets were analyzed for sample storage
effects (Table S5 in the Supplement).  It should however be noted that assessing the effect of
storage time on sample integrity was not a formal goal of the intercomparison exercise and



replicate samples were not analyzed at repeated intervals by independent laboratories, as would
normally be required for a thorough analysis. Nonetheless our results did provide some insights.
The comparison of measured concentrations (nmol kg$^{-1}$) and coefficients of variation (%) against
storage times revealed that low concentration methane samples (*i.e.* PAC1 and PAC2) were
susceptible to an increase in concentrations (Fig. 7a) and increased variability (Fig. 7b) with
increasing storage, as also observed by Magen et al. (2014). This was not as prevalent for higher
methane concentrations; however for samples with the highest methane concentrations *i.e.*
BAL5, there was some indication of a decrease in concentration with prolonged storage,
presumably as a result of gas leakage (Table S5 in the Supplement). For nitrous oxide, the
prominent observation was a potential decrease in concentration for higher concentration
seawater samples (*i.e.* BAL2), again presumably due to gas leakage. For low nitrous oxide
concentrations there was no comparable trend of increasing values to that observed for low
methane concentrations.

**4. Discussion**
The marine methane and nitrous oxide analytical community is growing. This is reflected in the
increasing number of corresponding scientific publications and the resulting development of a
global database for methane and nitrous oxide (Bange et al., 2009). Like all Earth observation
measurements, there is a need for intercomparison exercises of the type reported here, for data
quality assurance, and for appropriate reporting practices (National Research Council, 1993). To
the best of our knowledge, the work presented here is the first formal intercomparison of
dissolved methane and nitrous oxide measurements. Based on our results, we discuss the lessons
learned and our recommendations moving forward, by addressing the four questions that were
posed in the Introduction.

**4.1 What is the agreement between the SCOR gas standards and the 'in-house' gas**
**standards used by each laboratory?**
It is typical for laboratories to source some, or all, of their compressed gas standards from
commercial suppliers. National agencies, such as NOAA GMD or National Institute of
Metrology China, also provide standards to the scientific community. The national agencies
typically offer a lower range in concentrations than commercial suppliers, but their standards



tend to have a higher level of accuracy.  Of the twelve laboratories participating in the
intercomparison, eight reported using national agency standards, with seven of them using gases
sourced from NOAA GMD.  Since the methane and nitrous oxide mole fractions of these
national agency standards are equivalent to modern-day atmospheric mixing ratios, they are
similar to the SCOR ARS distributed to the majority of laboratories in this study.  Laboratories
in receipt of the SCOR standards were asked to predict their mole fractions based on those of
their own in-house standards.  For the majority that conducted this exercise, there was good
agreement (<3% difference) between the NOAA GMD and the SCOR ARS for both methane
and nitrous oxide.  For three laboratories, a larger offset was observed between the NOAA GMD
and the SCOR ARS.  There was also a good prediction for the higher methane content SCOR
WRS, facilitated by the linear response of the FID (Fig. 3).  In contrast, the nitrous oxide mole
fraction in the SCOR WRS exceeded the typical working range for several laboratories and it
was difficult for them to cross-compare with their in-house standards.  This reflects an analytical
set-up that involves on-column injection via a 6-port or 10-port valve with one or two sample
loops, respectively.  The sample loops have a fixed volume and their inaccessibility makes it
difficult to replace them by a smaller loop size.  Therefore either dilution of the standard is
required, or smaller loops need to be incorporated into the calibration protocol.  The two
laboratories that compared their in-house standards with the SCOR WRS reported an offset of
3% and >20%.  This indicates that variability between standards can be an issue for obtaining
accurate dissolved concentrations and provides support for the production of a widely available
high concentration nitrous oxide standard.  We strongly recommend that all commercially
obtained standards are cross-checked against primary standards, such as the SCOR ARS and
WRS.  This should be conducted at least at the beginning and end of their use to detect any drift
that may have occurred during their lifetime.   With due diligence and care, the SCOR standards
provide the capability for cross-checking personal standards for years to decades (Bullister et al.,

2016).


**4.2 How do measured values of methane and nitrous oxide compare across laboratories?**
**Methane:** The methane intercomparison highlighted the variability that exists between
measurements conducted by independent laboratories.  At low methane concentrations, a skewed
distribution of methane data was observed, which was particularly evident in PAC1 (Fig. 1a).





Potential causes include calibration procedures (Section 3.2) and/or sample contamination which
is more prevalent at low concentrations (Section 3.4). For some laboratories, the low methane
concentrations are close to their detection limit, which is determined by the relatively low
sensitivity of the FID and the small number of moles of methane in an introduced headspace
equilibration with seawater. An approximate working detection limit for methane analysis via
headspace equilibration is 1 nmol kg$^{-1}$, although some laboratories improve upon this by having
a large aqueous: gaseous phase ratio during the equilibration process (*e.g.* Upstill-Goddard et al.,
1996). Depending upon the volume of sample analyzed, purge-and-trap analysis can have a
detection limit much lower than 1 nmol kg$^{-1}$ (*e.g.* Wilson et al., 2017). Methane measurements
in aquatic habitats with methane concentrations near the limit of analytical detection include
mesopelagic and high latitude environments distal from coastal or benthic inputs (Rehder et al.,
1999; Kitidis et al., 2010; Fenwick et al., 2017). Of additional concern is that the skewed
distribution of methane concentrations also occurs in samples collected both from the surface
ocean (PAC2; Fig. 1b) and coastal environments (BAL1; Fig. 1c). Methane concentrations
between 2–6 nmol kg$^{-1}$ are within the detection limit of all participating laboratories. To address
this we recommend that laboratories restrict sample storage to the minimum time required to
analyze the samples and incorporate internal controls into their sample analysis (Section 4.4).
There was an improvement in the overall agreement between the laboratories for samples
with higher methane concentrations. However, some of the highest variability between the
laboratories was observed at the highest concentrations of methane analyzed (BAL2; Fig. 1e).
This high degree of variability resulted in significant uncertainty in the absolute *in situ*
concentration. Methane concentrations of this magnitude and higher are found in coastal
environments (Zhang et al., 2004; Jakobs et al., 2014; Borges et al., 2017) and in the water-
column associated with seafloor emissions (*e.g.* Pohlman et al., 2011). These environments are
considered vulnerable to climate induced changes and eutrophication, and therefore it is
necessary that independent measurements are conducted to the highest possible accuracy to
allow for inter-laboratory and inter-habitat comparisons. To address this we recommend that
reference material be produced and distributed between laboratories.

**Nitrous oxide**: Some of the trends discussed for methane were also evident in the nitrous oxide
data. For the samples with the lowest nitrous oxide concentrations a skewed data distribution





was observed, as found for methane (Fig. 4c). Such low nitrous oxide concentrations are typical
of low-oxygen water-column environments (<10 µmol kg$^{-1}$). Therefore, the analytical bias
towards measuring values higher than the absolute *in situ* concentrations is particularly pertinent
to oceanographers measuring nitrous oxide in oxygen minimum zones and other low-oxygen
environments (Naqvi et al., 2010; Farías et al., 2015; Ji et al., 2015). The low concentrations of
nitrous oxide still exceed detection limits by at least an order of magnitude for even the less-
sensitive headspace method due to the high sensitivity of the ECD. Therefore, the bias towards
reporting elevated values for low concentrations of nitrous oxide is related less to analytical
sensitivity and is more a consequence of calibration issues. During the intercomparison exercise
ECD calibration was identified as a nontrivial issue for all participating laboratories and it
deserves continuing attention. In particular, the nonlinearity of the ECD means that low and
high nitrous oxide concentrations are more vulnerable to error since the values fall outside of the
most frequented part of the calibration curve. This is particularly true if a linear fit is used to
calibrate the ECD (Fig. 6a). To circumvent this problem, one laboratory used a step-wise linear
function while other laboratories used a quadratic function. The usefulness of multiple
calibration curves for low and high nitrous oxide concentrations was highlighted during the
intercomparison exercise, although this necessitates some consideration of the threshold for
switching different calibration curves.

The majority of seawater samples analyzed had nitrous oxide concentrations ranging from 7–

11 nmol kg$^{-1}$ (Fig. 4a, 4b), which are close to atmospheric equilibrium values, as shown for the
Pacific Ocean (Fig. 4b). Collective analysis of these samples gives insight into the precision and
accuracy associated with surface water nitrous oxide analysis (Fig 5a). This is discussed further
in the context of implementing internal controls for methane and nitrous oxide (Section 4.4). For
samples with the highest nitrous oxide concentrations, *i.e.* exceeding 30 nmol kg$^{-1}$, there was
high variability between the concentrations reported by the independent laboratories. This was
most evident for the BAL5 samples (Fig. 4e) and similar to the variability observed at the highest
methane concentrations analyzed (Fig. 1e). It is difficult to assess how much of this variability
was specifically due to the differences in calibration practices between the laboratories and the
differences in gas standards with high nitrous oxide mole fractions, but at least some of it can be
attributed to this. These results form the basis for a proposed production of reference material
for both trace gases.




**4.3 Are there general recommendations to reduce uncertainty in the accuracy and precision of methane and nitrous oxide measurements?**

Lessons learned during the intercomparison exercises will be the basis for a forthcoming Good Practice Guide for dissolved methane and nitrous oxide. Key points include the use of traceable gas standards (discussed in Section 4.1), calibration fits (discussed in Section 3.2 and 3.3), sample storage time (discussed in Section 3.4), internal controls, and reference material.

Laboratories participating in this intercomparison exercise used one of two analytical approaches; either headspace equilibration or the purge-and-trap technique. Aside from the low methane concentrations, for some laboratories both analytical approaches yielded comparable values for methane and nitrous oxide. At sub-nanomolar methane concentrations, four out of the six laboratories that reported methane concentrations <1 nmol kg$^{-1}$ used a purge-and-trap analysis.

Internal controls represent a self-assessment quality control check to validate the analytical method and quantify the magnitude of uncertainty. Appropriate internal controls for methane and nitrous oxide consist of air-equilibrated seawater samples. Their purpose is to provide checks for methane concentrations ranging from 2–3 nmol kg$^{-1}$ and for nitrous oxide concentrations from 5–9 nmol kg$^{-1}$. The air used in the equilibration process could derive from the ambient environment if sufficiently stable or from a compressed gas cylinder after cross-checking the concentration with the appropriate gas standard. Air-equilibrated samples provide reassurance that the analytical system is providing values within the correct range. Air-equilibrated samples also indicate the certainty associated with calculating the saturation state of the ocean with respect to atmospheric equilibrium. This is particularly relevant when the seawater being sampled is within a few percent of saturation. Finally, these air-equilibrated samples provide an estimate of analytical accuracy, which is infrequently reported for methane or nitrous oxide. At present, only a few studies report the analysis of air-equilibrated seawater alongside water-column samples (Bullister and Wisegarver, 2008; Capelle et al., 2015; Wilson et al., 2017). It is considered likely that wider implementation would facilitate internal assessment of the analytical system. Since the main equipment required is a water-bath and an overhead stirrer, the production is not cost-prohibitive. A recommendation of this intercomparison



exercise is that laboratories routinely use air-equilibrated seawater samples to provide an
estimate of analytical accuracy.
In addition to the self-assessments provided by the analysis of air-equilibrated seawater, this
study revealed the need for reference seawater to help assess the accuracy of high concentration
methane and nitrous oxide measurements. Reference seawater in this instance refers to batches
of dissolved methane and nitrous oxide samples prepared in the laboratory using an equilibrator
set-up, as used for dissolved inorganic carbon (Dickson et al., 2007). In the absence of plans for
additional intercomparison exercises, the provision of reference seawater will allow laboratories
to continue evaluating their own measurements.

**599 4.4 What are the implications of interlaboratory differences for determining the spatial and**

**600 temporal variability of methane and nitrous oxide in the oceans?**

The key outcome of this study was the identification of differences in methane and nitrous oxide
concentrations for the same batch of seawater samples measured by several independent
laboratories. Emergent from this is the distinct possibility that any given laboratory will
incorrectly report data, thereby increasing uncertainty over the saturation states of both gases.
The tendency to over-estimate methane concentrations close to atmospheric equilibrium means
that marine emissions of methane to the overlying atmosphere will be also overestimated (Bange
et al., 1994; Upstill-Goddard and Barnes, 2016). In contrast, for nitrous oxide there does not
appear to be either an under-estimation or over-estimation of concentrations. Consequently, there
is generally a lower inherent uncertainty in its surface ocean saturation state, as previously
proposed (Law and Ling, 2001; Forster et al., 2009).
The inter-laboratory differences highlighted by this study should be viewed in the context of
numerous individual efforts to assess temporal and/or spatial trends in methane and nitrous oxide
by way of time-series observations (Bange et al., 2010; Farías et al., 2015; Wilson et al., 2017;
Fenwick and Tortell, 2018), repeat hydrographic survey lines (de la Paz et al., 2017), and single
expeditions. While the value of these in integrating the behaviour of methane and nitrous oxide
into the hydrography and biogeochemistry of local-regional ecosystems is beyond question, their
value would be enhanced by the rigorous cross-validation of analytical protocols. Without this,
perceived small temporal and/or spatial changes in water-column concentrations in any given
region are difficult to verify unless the data all originate from a single laboratory. In addition,



the value of a global methane and nitrous oxide database (*e.g* Bange et al., 2009) would to some
extent be compromised by the uncertainty. Taking due account of the analytical variability
between laboratories will clearly be vital to any future assessment of the changing methane and
nitrous oxide budgets of the oceans.

**5. Conclusions**
Overall, the intercomparison exercise was invaluable to the growing community of oceanic
methane and nitrous oxide analysts.  The level of agreement between independent measurements
of dissolved concentrations was evaluated in the context of several contributing factors,
including sample analysis, standards, calibration procedures, and sample storage time.
Importantly, the intercomparison represents a concerted effort from the scientists involved to
critically assess the quality of their data, and to initiate the steps required for further
improvements.  Recommendations arising from the intercomparison include routine cross-
calibration of working gas standards against primary standards, minimizing sample storage time,
incorporating internal controls (air-equilibrated seawater) alongside routine sample analysis, and
the future production of reference seawater for methane and nitrous oxide measurements.  These
efforts will help resolve temporal and spatial variability, which is neccesary for constraining
methane and nitrous oxide emissions from aquatic ecosystems and for evaluating the processes
that govern their production and consumption in the water-column.



*Acknowledgements:* The methane and nitrous oxide intercomparison exercise was conducted as a
Scientific Committee on Ocean Research (SCOR) Working Group which receives funding from
the U.S. National Science Foundation (OCE-1546580).  Pacific Ocean seawater samples were
collected on HOT cruises which are supported by NSF (including the most recent OCE-1260164
to DMK).  Baltic Sea seawater samples were collected during Cruise #142 of the RV Elisabeth
Mann Borgese, with the ship-time provided by the Leibniz Institute for Baltic Sea Research
Warnemünde.  We thank Liguo Guo for help with sampling during the Baltic Sea cruise.  The
methane and nitrous oxide gas standards were produced via a Memorandum of Understanding
between the University of Hawaii and NOAA-PMEL.  Funding for the gas standards was
provided by for the Center for Microbial Oceanography: Research and Education (C-MORE;
EF0424599 to DMK), SCOR, the EU FP7 funded Integrated non-$CO_2$ Greenhouse gas
Observation System (InGOS) (Grant Agreement #284274), and NOAA's Climate Program
Office, Climate Observations Division.  Additional support was provided by the Gordon and
Betty Moore Foundation #3794 (DMK), the Simons Collaboration on Ocean Processes and
Ecology (SCOPE; #329108 to DMK), and the Global Research Laboratory Program (#
2013K1A1A2A02078278 to DMK) through the National Research Foundation of Korea (NRF).
AVB is a senior research associate at the FRS-FNRS.  AES would like to acknowledge NSF
OCE-1437310. MP would like to acknowledge the support of the Spanish Ministry of Economy
and Competitiveness (CTM2015-74510-JIN).  Any use of trade names is for descriptive
purposes and does not imply endorsement by the U.S. government




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



**Table 1.** List of laboratories that participated in the intercomparison. All laboratories measured
both methane and nitrous oxide except U.S. Geological Survey (methane only), UCSB (nitrous
oxide only), and NOAA PMEL (nitrous oxide from the Pacific Ocean). Also indicated are the
twelve laboratories that received the SCOR gas standards of methane and nitrous oxide.

| Institution | Lead Scientist | SCOR Standards |
|---|---|---|
| University of Hawaii, USA | Samuel Wilson | Yes |
| GEOMAR, Germany | Hermann Bange | Yes |
| Newcastle University, UK | Robert Upstill-Goddard | Yes |
| Université de Liège, Belgium | Alberto Vieira Borges | No |
| Plymouth Marine Laboratory, UK | Andrew Rees | Yes |
| NOAA PMEL, USA | John Bullister | Yes |
| IIM-CSIC, Spain | Mercedes de la Paz | Yes |
| CACYTMAR, Spain | Macarena Burgos | No |
| University of Concepción, Chile | Laura Farías | Yes |
| IOW, Germany | Gregor Rehder | Yes |
| University of California Santa Barbara, USA | Alyson Santoro | Yes |
| National Institute of Water and Atmospheric Research, NZ | Cliff Law | Yes |
| University British Columbia, Canada | Philippe Tortell | Yes |
| USGS, USA | John Pohlman | No |
| Ocean University of China, China | Guiling Zhang | Yes |






**Table 2.** Pertinent information for each batch of methane and nitrous oxide samples. This
includes contextual hydrographic information, median and mean concentrations of methane and
nitrous oxide, range, number of outliers, and the overall average coefficient of variation (%).

| Sampling parameters | | | | | | | | | |
|---|---|---|---|---|---|---|---|---|---|
| Sample ID | PAC1 | PAC 2 | BAL1 | BAL2 | BAL3 | BAL4 | BAL5 | BAL6 | BAL7 |
| Location | 22.75N 158.00W | 22.75N 158.00W | 54.32N 11.55E | 54.11N 11.18E | 55.25N 15.98E | 55.30N 15.80E | 55.30N 15.80E | 54.47N 12.21E | 54.47N 12.21E |
| Location name | Station ALOHA | Station ALOHA | TF012 | TF022 | TF213 | TF212 | TF212 | TF046a | TF046a |
| Sampling date | 24.2.17 | 24.2.17 | 16.10.16 | 17.10.16 | 18.10.16 | 19.10.16 | 20.10.16 | 21.10.16 | 21.10.16 |
| Sampling depth (m) | 25 | 700 | 3 | 22 | 3 | 92 | 71 | 3 | 21 |
| Seawater temperature ($^o$C) | 23.6 | 5.1 | 12.0 | 13.6 | 12.2 | 6.6 | 6.7 | 11.8 | 13.4 |
| Salinity | 34.97 | 34.23 | 13.85 | 17.37 | 7.87 | 18.40 | 18.08 | 8.81 | 17.65 |
| Density (kg m$^{-3}$) | 1024 | 1027 | 1010 | 1013 | 1006 | 1014 | 1014 | 1006 | 1013 |
| **Nitrous oxide** | | | | | | | | | |
| Number of datasets | 13 | 13 | 12 | 13 | 12 | 13 | 12 | 13 | 12 |
| Outliers | 0 | 1 | 2 | 1 | 1 | 0 | 1 | 2 | 2 |
| Median N$_2$O conc. (nmol kg$^{-1}$) | 42.4 | 7.0 | 11.0 | 9.4 | 11.1 | 3.4 | 40.2 | 11.0 | 9.6 |
| Mean N$_2$O conc. (nmol kg$^{-1}$) | 41.3 | 7.0 | 11.1 | 9.2 | 11.0 | 3.4 | 39.0 | 10.8 | 9.5 |
| Range | 34.3-45.8 | 5.9-7.6 | 10.1-12.7 | 7.7-11.0 | 9.6-11.6 | 2.1-5.5 | 30.1-45.9 | 9.5-11.5 | 8.0-10.4 |
| Average coeff. variation (%) | 2.8 | 4.4 | 4.5 | 4.2 | 2.7 | 7.5 | 4.0 | 2.6 | 4.4 |
| **Methane** | | | | | | | | | |
| Number of datasets | 12 | 12 | 11 | 11 | 11 | 11 | 11 | 11 | 11 |
| Outliers | 0 | 1 | 0 | 0 | 0 | 1 | 1 | 0 | 0 |
| Median CH$_4$ conc. (nmol kg$^{-1}$) | 0.9 | 2.3 | 5.7 | 60.3 | 4.1 | 31.3 | 18.8 | 5.0 | 35.2 |
| Mean CH$_4$ conc. (nmol kg$^{-1}$) | 1.8 | 2.6 | 5.8 | 58.6 | 4.4 | 31.1 | 18.8 | 5.4 | 35.4 |
| Range | 0.6-3.1 | 1.9-3.8 | 2.9-8.9 | 45.2-67.2 | 2.5-6.5 | 26.9-35.3 | 16.5-20.7 | 3.8-6.8 | 30.1-42.1 |
| Average coeff. variation (%) | 10.9 | 7.2 | 8.6 | 2.1 | 4.3 | 3.5 | 4.2 | 6.5 | 3.5 |





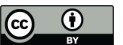

**Figures**

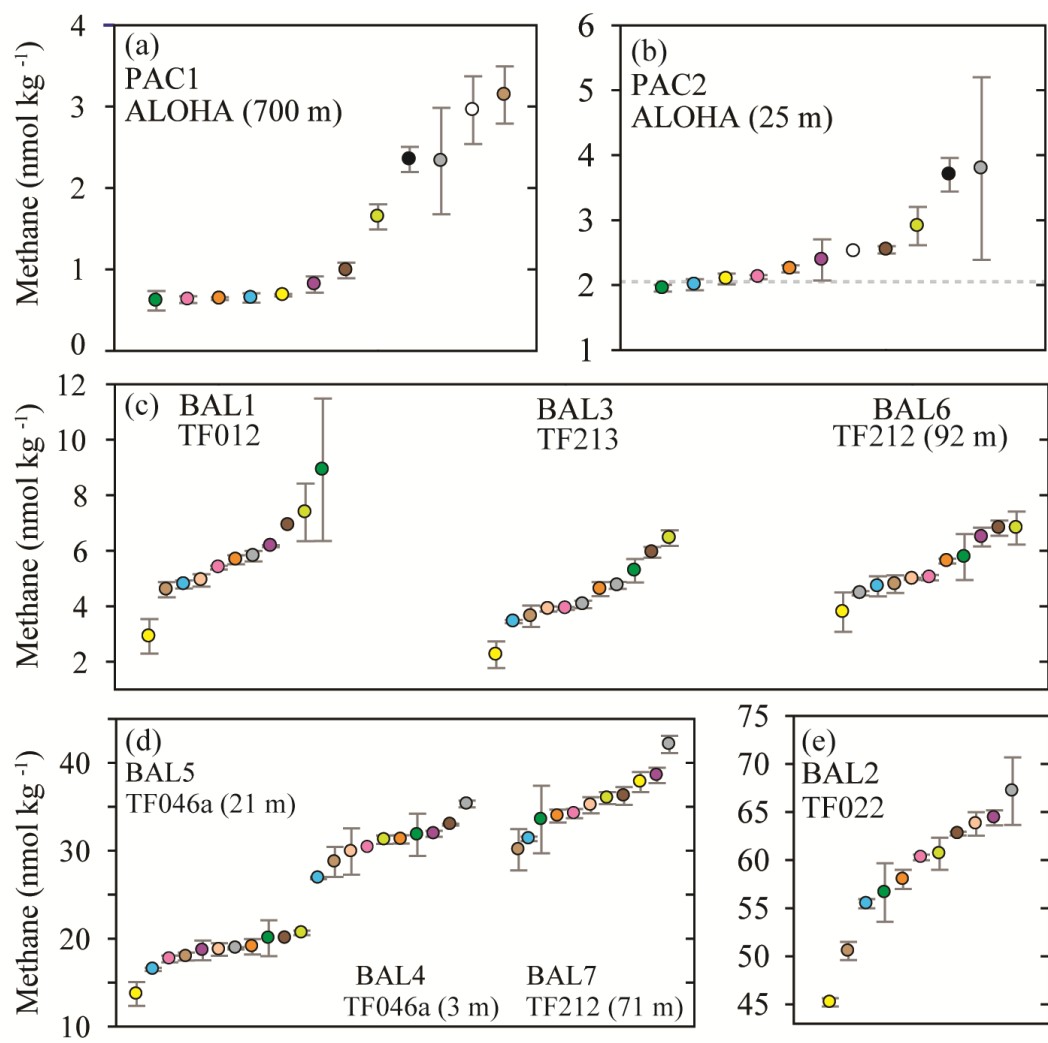

Figure 1. Concentrations of methane measured in nine separate seawater samples collected from
the Pacific Ocean (Fig. 1a, 1b) and the Baltic Sea (Fig. 1c, 1d, 1e). The dashed grey line
represents the value of methane at atmospheric equilibrium (Fig. 1b.) Individual data points are
plotted sequentially in increasing value with the same color symbol for each laboratory in all
plots.



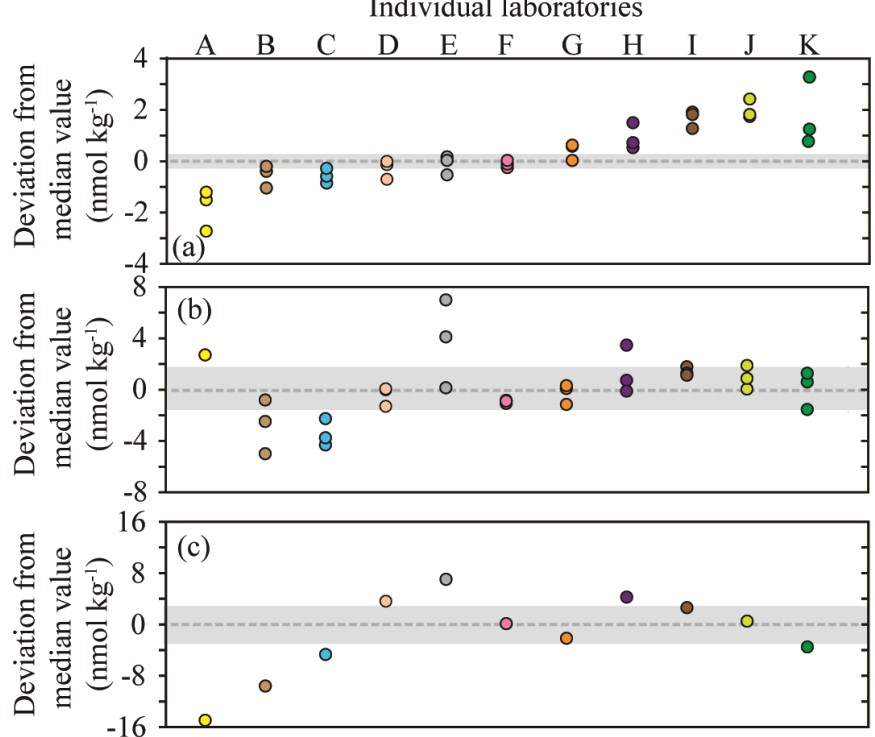

Figure 2. Deviation from the median methane concentration (reported as absolute values in nmol
kg$^{-1}$) for the seven Baltic Sea samples. The batches of seawater samples include BAL1, BAL3,
and BAL6 (Fig. 2a), BAL4, BAL5, and BAL7 (Fig. 2b), and BAL2 (Fig. 2c). The shaded grey
area indicates values ≤5% of the median concentration. The color scheme for each laboratory
dataset is identical to that used in Figure 1 and the letters allocated to each dataset are to facilitate
cross-referencing in the text. Note that the y-axis scale varies between the Figures.






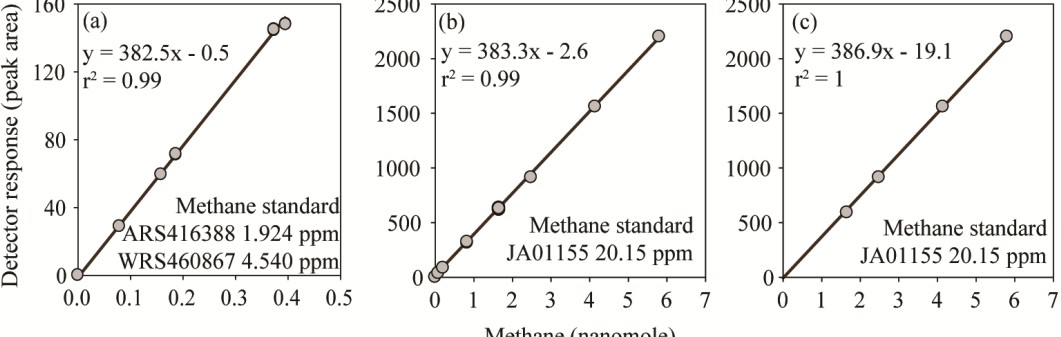


Figure 3. FID response to methane, fitted with a linear regression calibration.  The inclusion
(Fig. 3a and Fig. 3b) or exclusion (Fig. 3c) of low methane values cause the calibration slope and
intercept to vary.  However, the observed variation in the calibration slope does not have a
significant effect on the final calculated concentrations of methane.  In contrast, variation in the
intercept does have an effect on the final concentrations of methane.



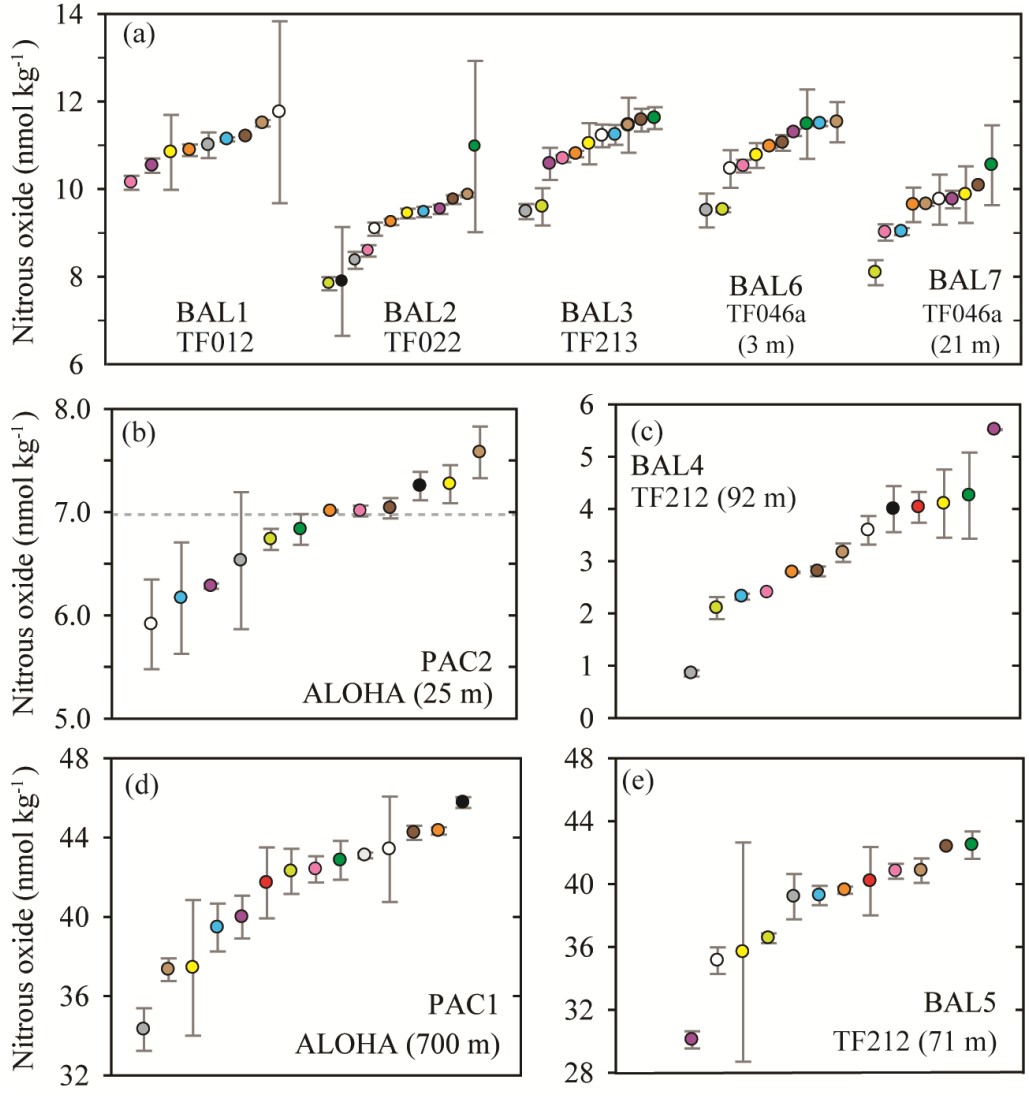

Figure 4. Concentrations of nitrous oxide measured in nine separate samples from the Baltic Sea
and the Pacific Ocean. The dashed grey line represents the value of nitrous oxide at atmospheric
equilibrium (Fig. 4b). Individual data points are plotted sequentially in increasing value with the
same color symbol for each laboratory in all plots.





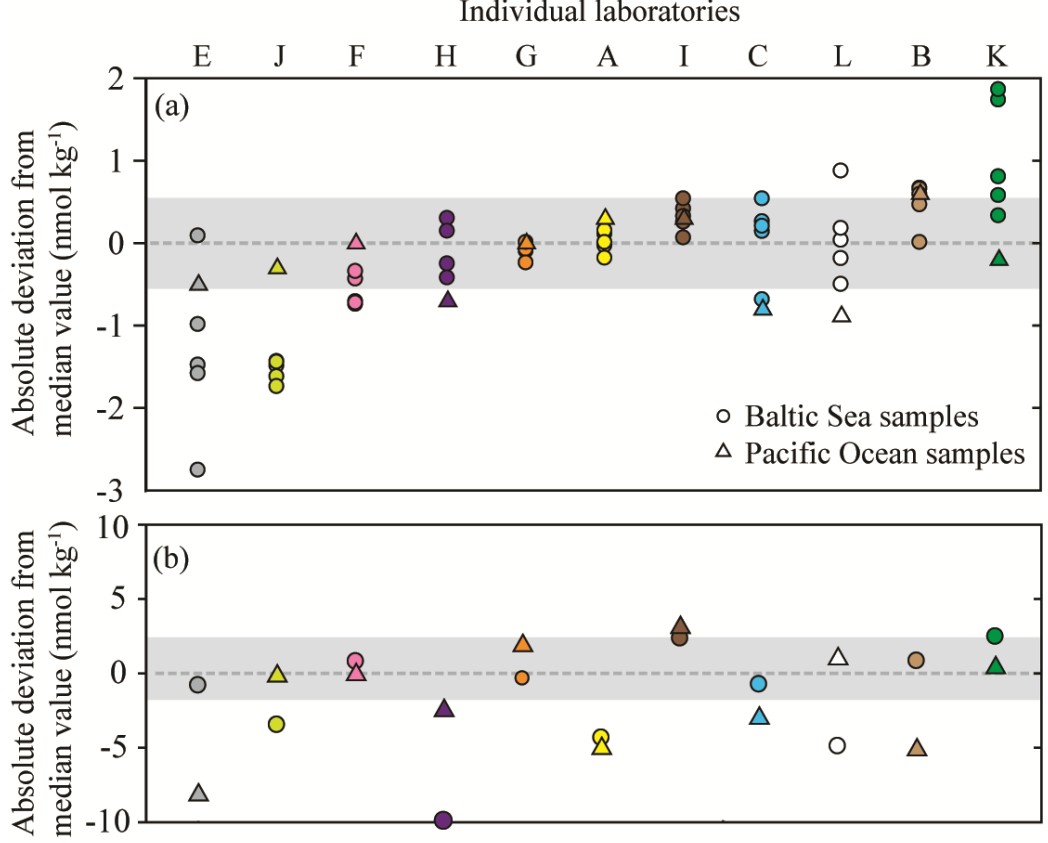


Figure 5. Deviation from the median value (reported in absolute units) for nitrous oxide datasets.
The batches of samples include BAL1,2,3,6,7 (Fig. 5a) and PAC2 and BAL5 (Fig. 5b). The
Baltic Sea samples are represented by circles and the Pacific Ocean samples are represented by
triangles. The shaded area indicates values ≤5% based on a water-column concentration of 11
nmol kg$^{-1}$ and 42 nmol kg$^{-1}$ for Fig. 5a and 5b, respectively. The color scheme for each
laboratory dataset is identical to that used in Figure 4 and the letters allocated to each dataset are
to facilitate cross-referencing in the text. Note the y-axis for Fig 5a and 5b are plotted on a
different scale.



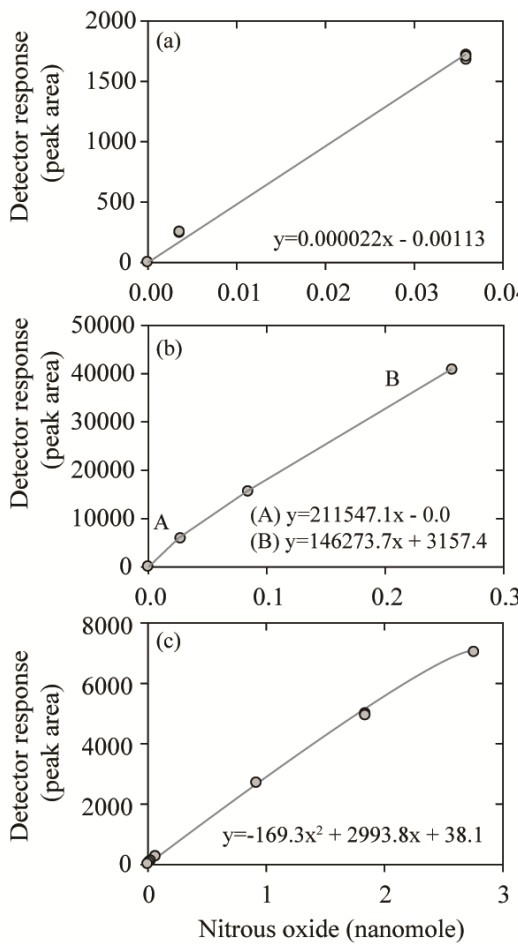


Figure 6. Three calibrations curves for nitrous oxide measurements using an ECD including
linear (Fig. 6a), multilinear (Fig. 6b), and quadratic (Fig. 6c).






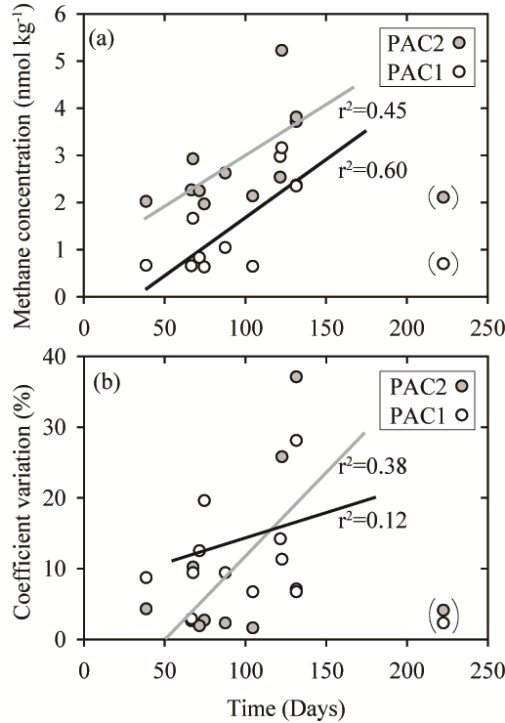


Figure 7. Comparison of sample storage times with measured concentrations of methane (Fig.
7a) and coefficient variation (Fig. 7b) for two sets of seawater samples (PAC1 and PAC2).
These two sets of seawater samples had the lowest methane concentrations and appear to be
influenced by the duration of storage time. The data points enclosed in parentheses were not
included in the regression analysis. The PAC1 regression line is black and the PAC2 regression
line is grey. All of the storage times are included in the Supplementary Material.