# Peer review of "An intercomparison of oceanic methane and nitrous oxide measurements"

_Biogeosciences, 2018_

## Referee Comment (RC1) · Anonymous Referee #1 · 18 Jul 2018

**General comments:**

Wilson et al. present the first intercomparison of oceanic methane and nitrous oxide measurements across numerous (n = 11) international laboratories. This is a timely and important contribution for the community. The paper is scientifically sound, well-written and clear. I have few (generally minor) comments/suggestions below.

While this intercomparison is a first step toward being able to compare the concentrations of these gases measured by different laboratories in marine environments, I have some recommendations to improve the paper. First, while they could discern some trends, I don't think the effect of storage can easily be isolated if the samples are not collected the same way (e.g., using same vial sizes, stoppers) and analyzed using the same analytical method. Although admittedly not being the focus of the present paper, a storage experiment should be repeated where samples in each dataset would be sequentially analyzed at different time points by the same laboratory (all other things being equal). Different type of stoppers/seals should also be compared to determine which one is best.

Also, because water budgets are often limited, they should better assess the effect of different sample volumes on precision and exactitude if possible. For instance, are samples with larger volumes yielded better results?

**Minor comments**

Page 4, lines 85-89: Which method is the most sensitive (purge and trap versus headspace equilibration)? Discuss the advantages/inconveniences of using one over the other a bit more.

Page 6, lines 140-156: The part describing how they determined the absolute mole fractions for these standards is not clear and the link (www.scor-int.org /SCOR_Publications) is not working. Why would the uncertainty be higher for the nitrous oxide WRS standard compared to the methane one?

Page 7, lines 158- 182: The effects of sample volumes, type of septa used and storage should be assessed better since these differed between the laboratories involved in the intercomparison.

Page 7, line 171-173: Was there a difference between sampling bottles?

Page 7, line 178: Which kind of stopper? Also, what is the effect of different stoppers/seals used during storage? Are some stoppers/seals leaking more than others?

Page 7, lines 180-182: They used mercuric chloride for preservation, which is probably acceptable for water-column samples. However, mercuric chloride is toxic and difficult to ship and use at sea due to safety concerns. Future efforts should test alternative types of preservatives (sodium hydroxide, formaldehyde) to evaluate their suitability to preserve these samples in different marine environments. Also, mercuric chloride might not be suitable for some marine samples as Ostrom et al (2016) suggest that it could enhance nitrous oxide production by chemodenitrification in Fe-rich environments.

Page 8, line 188: I assume this tank was gas tight?

Page 8, lines 196-198: Was there a difference between this first and last samples? Any change in temperature during sampling would affect gas concentrations. Also, I suppose a headspace was created in the 1000 L water tank as samples were drawn?

Page 9, lines 223-225: "headspace collected into a gas tight syringe and injected": How is this different than the physical injection?

Page 9, lines 228-229: How many standards were typically used?

Page 9, line 248: Why does the tubing need to be maintained at low temperatures?

Page 9, line 249: Low blank for what? Methane, nitrous oxide, or both?

Page 10, line 251-252: Be more specific: "liquid nitrogen (-165$^{o}$C) for nitrous oxide or cooled ethanol (-70$^{o}$C) for methane."

Page 11, line 303: By "comparable values" do you mean peak area?

Page 13, lines 362-371: This point comes across more clearly in the Fig. 3's legend. Perhaps rewrite?

Page 14, lines 388-401: A sample with higher nitrous oxide concentrations could also be used in future intercomparison efforts. For instance, nitrous oxide concentrations of up to 1000 nmol/L were measured in coastal waters off Peru (Arévalo-MartÍnez et al.,2013)..

Page 15: Why was the variability higher for the BAL5 dataset? Could this be related to sampling and/or storage?

Page 16, lines 438-439: Was this only true for samples with methane concentrations less than atmospheric concentrations?

Page 18, line 512: What would be their maximum recommended storage time?

Page 19, lines 532-534: They discuss detection limits for methane but not for nitrous oxide analysis methods. What are the detection limits associated with the two different analysis methods (headspace equilibration versus purge and trap)?

Page 20, lines 560-565: Other important points, e.g., sample volume, septa/seals used, preservative used, should also be included in future efforts.

Page 20, line 576-577: This assumes that the air in the laboratory where the measurements are done is not contaminated by other sources of nitrous oxide (non-atmospheric).

Page 20, line 586: Bourbonnais et al. (2017) also used air-equilibrated seawater standards to calculate water-column nitrous oxide concentrations off Peru.

**Figures:**

Figures 1: Are values of methane at atmospheric equilibrium expected at 25 m depth? Is this in the mixed layer?

Figure 7: Are these relationships significant (add $r^2$)? Ideally, to assess storage effects, samples collected the same way and using the same analysis method should be analyzed at different time points by the same laboratory.

**Supplementary Materials:**

Tables 6 and 7: Add detection limits for each laboratory.
Add last name "Macarena Burgos" as done for all other researchers.

**Few technical points**

Page 4, lines 76 to 78: Typically is used twice in these two sentences – remove one instance.

Page 18, line 501: change "equilibration" for "equilibrated".

Page 19, line 545: change to "switching between different calibration curves."

**Additional references:**

Bourbonnais, A., Letscher, R.T., Bange, H.W., Echevin, V., Larkum, J., Mohn, J., Yoshida, N. and Altabet, M.A., 2017. N2O production and consumption from stable isotopic and concentration data in the Peruvian coastal upwelling system. *Global Biogeochemical Cycles*, *31*(4), pp.678-698.

Ostrom, N.E., Gandhi, H., Trubl, G. and Murray, A.E., 2016. Chemodenitrification in the cryoecosystem of Lake Vida, Victoria Valley, Antarctica. *Geobiology*, *14*(6), pp.575-587.

---

## Referee Comment (RC2) · Anonymous Referee #2 · 30 Jul 2018

In their manuscript, Wilson et al. present data from a recent international intercomparison study which evaluated the analytical procedures used to measure the concentrations of methane and nitrous oxide dissolved in seawater. Specifically, seawater samples and gaseous standards were sent to several different laboratories for analysis. Since the measurement of methane and nitrous oxide concentrations are mainly done in the gas, not liquid, phase, the different laboratories had different protocols to first separate the dissolved gas prior to analysis as well as the final analysis; while the different labs had different protocols, they mainly involved either headspace equilibration or a purge and trap technique.

The results of this intercomparison are striking, with different laboratories reporting concentrations that could be different by several hundred percent. The highest per-

cent differences were reported for the lowest concentration samples, and since low concentrations are typically reported in the near-surface waters, this inter-laboratory difference is particularly troubling for global extrapolation of sea-to-air fluxes for these two gases. The impact of this manuscript is that it identifies significant inconsistencies between laboratories, and while the data from any one laboratory is likely valid for testing hypotheses, combining data from multiple laboratories for global extrapolation or time series analysis will lead to significant unknowns.

A the end of the manuscript, the reader is left hungry for more, wondering how these inconsistencies might be rectified with a hypothetical Standard Operating Procedure. But while the authors provide a few recommendations for how to lower uncertainties, they do not prove the major cause of these inconsistencies, and thus which procedure might be preferred. The authors appropriately did not attempt this recommendation as it was beyond what their data can illuminate. For example, a full analysis of the headspace equilibration procedure would require each laboratory to establish the accuracy and precision of each variable in Equation 1 (pressure, temperature, salinity, headspace volume, and water volume) using their procedures. The authors assess the calibration of the analytical instrument and the variability of the overall results, but not these specific variables. In addition, the authors recognize that storage time is a variable significantly influencing the results. Since these additional variables were not systematically investigated, the authors are correct in not recommending a preferred procedure, and instead choose to report overall inconsistencies.

Sample storage: I recommend that the authors expand section 3.4. I found this section too brief on experimental details and I was left assuming how storage time was assessed. Was the sample storage time variable controlled in any systemic way or is this simply the time it took different labs to actually conduct their analyses? Is there any way to normalize the data in Figures 1 and 4 to sample storage time or would that be extending this data too far? Can the authors assess how much variation in the dissolved concentrations is due to storage vs. procedure?

[Figure]

The authors suggest that leakage may be a source of uncertainty for longer storage times, but they don't raise the possibility of inadequate preservation. Most groups analyzing these dissolved gases assume that adding enough mercuric chloride to a sample will halt all biological activity, but that may not be the case. In addition, what is the chance that gases are outgassing or adsorbing to the stopper? Since these are both possible influences on the final results, I suggest that the authors also briefly raise these possibilities.

Overall, this investigation appears robust and the manuscript is well written. The authors have uncovered a significant result which will benefit the community.

---

## Referee Comment (RC3) · Anonymous Referee #3 · 13 Aug 2018

The authors present a very important result of an intercomparison between many labs for measuring methane and nitrous oxide levels in ocean water samples. Overall, I think this paper is well written and will be a great contribution to the field. A lot of planning and work went into this study, and is worthy of publishing. The main focus is to look at standards, calibration issues, but don't really address how with the large variability of how people process water samples affects the results. I think this paper highlights some very important issues regarding trace gas analysis in open ocean settings, and could be transferred to other environments. Section 4.3 will be regarded as a huge step forward, once this group is able to produce a Good Practice Guide to the community. While I was left wanting to know about how best to make these measurements, I acknowledge that this group is on the way to doing that and will do that. This paper is the first step. The conclusion that calibration issues are a huge problem in this field, and the recommendation to produce reference material for both trace gases is a wonderful contribution.

1. They mention on line 587 for all labs to do internal checks by measuring an air-equilibrated seawater. They mention needing a water bath and stirrer. Since this is a main finding that could be implemented in the community ASAP, could they provide true details of the setup? This might be appropriate in the supplementary materials.

2. Line 220: Why is there such variation in equilibration time for the gases; between 20 min to 24 hours? Has anyone done a time series of equilibration times to show what the time needs to be? This could be part of the recommendations.

3. Ling 272: Where do the CV values come from that are plotted in figure 7b? In table S2, there is one column for "mean CV" which seems to be related to each lab, and not specifically for PAC1 and PAC2. Maybe those CVs are just not reported in the table, in which case, please report them.

4. Line 371: it is not clear to me what they mean by "sample contamination, discussed below (datasets J and K)." Where do they discuss below? Could they call out the specific sections they want the reader to refer to?

5. Line 430 and on: The storage section really added a nice dimension to the paper, even though it was not a main focus. On line 445, you state that BAL2 shows a decrease in N2O concentrations over time. Can you show that graph? When graphed, I see that BAL2 shows an increase with time but it also seems within the variability of the measurements.

[Figure]

6. Line 432: The explanation of the results from Magen 2014 are a bit misleading. That paper shows that at methane concentrations less than ~1ppm in the headspace, there could be a storage issue after 1 year. And the issue is that concentrations increase. There should be more

context to your statement "because prolonged sample storage adversely affects dissolved methane and nitrous oxide samples (Magen et al., 2014)…."

7. Line 439: Storage for methane. Where did the data come from for figure 7? From the supplemental tables, the only storage time data shown is from Feb for PAC 2, and Nov for PAC1. Just from a first look, there are only 7 reported values for methane for PAC1 Nov in Table S2, but 11 points plotted in figure 7a. Where is the extra data coming from? Data looks consistent for PAC2. If I replot the storage days from table S5 vs the concentrations from table S2, I get the following graphs. (For the graphs below, methane concentrations are plotted over storage time with the outliers and without.) Those outliers were identified in figure 7a with () around the symbols, which is stated in the figure cation to be taking out of the regression. For PAC2, I reproduce what was reported in figure 7a, but for PAC1, the story is completely different. Please address this inconsistency.

   a. After agonizing over the mismatch of this data, it looks like they plotted PAC1 Feb 2017 in figure 7a, not PAC1 Nov 2013. If that's the case, the storage time data presented in table S5 is not right.

[Figure]

8. Can you add a column in the supplemental table for N2O for how each person dealt with water, like what was done for methane? Water is a huge issue for N2O precision, and there is no mention of how water was dealt with.
9. Line 507, if your intent is to show some examples, you should add "for example" to your reference list here. There are many other papers that show this.
10. Line 557: extra space between "proposed" and "production"
11. In table S5, "red" is listed as having measured something on the PAC samples 140 days after collection. But when I try to cross reference this in table 2, it looks like "red" didn't measure for methane. It might help to know if the storage times in table S5 are for methane and/or N2O. Overall, I think this table needed revisiting.
12. Figure S1, what is the gray dashed line? What do colors represent?
13. Figure S2, are a and b shallow water and c and d deep water? Make that clear in the first description of the figure. It says "same location" but what you mean is at the same lat/long but two different depths. Also, caption says "In contrast, the concentration of nitrous oxide in the deep-water samples (Figure S2c and d) was more consistent and the data values for the laboratories that measured samples from 2013 and 2017 are shown together in Figure S2d." is that also supposed to be shown by a gray dashed line? Can you make the scales the same for both sides?
14. Supp table 1: what is the point of the far right columns in this table? What is the mean CV of? For example, for lab A, it says 9.2% CV. Did you take CV for each BAL1, BAL2, etc, and then average that? Since we don't see the BAL1 CV, this is not clear. That being said, I'd like to see the CV for the standards run in the lab. From my experience with N2O, I can have ~10% CV if there is still water in the sample.

---

## Author Comment (AC1) · 31 Aug 2018

We would like to thank the three Reviewers for their detailed comments, which have improved the manuscript. We have included a point-by-point response to their comments below, with our responses highlighted in bold text.

Reviewer #1 General comments: Wilson et al. present the first intercomparison of oceanic methane and nitrous oxide measurements across numerous (n = 11) international laboratories. This is a timely and important contribution for the community. The paper is scientifically sound, well-written and clear. I have few (generally minor) comments/suggestions below. While this intercomparison is a first step toward being able to compare the concentrations of these gases measured by different laboratories in

marine environments, I have some recommendations to improve the paper. First, while they could discern some trends, I don't think the effect of storage can easily be isolated if the samples are not collected the same way (e.g., using same vial sizes, stoppers) and analyzed using the same analytical method. Although admittedly not being the focus of the present paper, a storage experiment should be repeated where samples in each dataset would be sequentially analyzed at different time points by the same laboratory (all other things being equal). Different type of stoppers/seals should also be compared to determine which one is best. Also, because water budgets are often limited, they should better assess the effect of different sample volumes on precision and exactitude if possible. For instance, are samples with larger volumes yielded better results? Thank you for these overall positive comments. We address the issue of storage artifacts below.

Minor comments Page 4, lines 85-89: Which method is the most sensitive (purge and trap versus headspace equilibration)? Discuss the advantages/inconveniences of using one over the other a bit more. We have updated the text in the Introduction and Lines 89-95 now read 'The purge and trap technique is typically more sensitive by 2-3 orders of magnitude over headspace equilibrium. However, the purge and trap technique requires more time for sample analysis and it is more difficult to automate the injection of samples into the gas analyzer. Headspace equilibrium sampling is most suited for volatile compounds that can be efficiently partitioned into the headspace gas volume from the seawater sample. Its limited sensitivity can be compensated by large volume analysis (e.g. Upstill-Goddard et al., 1996).' The different merits of the two methods are also featured in the revised Discussion, where we highlight the detection limits for methane which are more of an issue than for nitrous oxide. Lines 519-522 read 'An approximate working detection limit for methane analysis via headspace equilibration is 1 nmol kg-1, although some laboratories improve upon this by having a large aqueous: gaseous phase ratio during the equilibration process (e.g. Upstill-Goddard et al., 1996). Depending upon the volume of sample analyzed, purge-and-trap analysis can have a detection limit much lower than 1 nmol kg-1 (e.g. Wilson et al., 2017).'

Page 6, lines 140-156: The part describing how they determined the absolute mole fractions for these standards is not clear and the link (www.scor-int.org /SCOR_Publications) is not working. Why would the uncertainty be higher for the nitrous oxide WRS standard compared to the methane one? We apologize that the report which documented the production of the gas standards was not easily accessible. It is now accessible through the University of Delaware library and the citable URI is now included in the appropriate reference (http://udspace.udel.edu/handle/19716/23288). The report is also attached to this response for your convenience. On Pages 4-5 of this report, the calibrations for the nitrous oxide and methane WRS are described. In response to the question, there is higher certainty for the ARS because the standards were cross-calibrated with National Oceanic and Atmospheric Administration/Climate Monitoring and Diagnostics Laboratory (NOAA/CMDL) and Advanced Global Atmospheric Gases Experiment (AGAGE) standards which have a similar mole fraction. In contrast, the mole fraction of the nitrous oxide WRS far exceeds that of the CMDL and AGAGE standards and the calibration curves are highly non-linear. Therefore, the reported 2-3% accuracy takes into consideration the likelihood of increased systematic errors.

Page 7, lines 158- 182: The effects of sample volumes, type of septa used and storage should be assessed better since these differed between the laboratories involved in the intercomparison. Reviewer #1 points out that there were sampling and storage variables which were not controlled for during the intercomparison exercise. These are responded to separately below

Sample bottle size We have taken the Reviewer's comments into consideration and expanded Section 3.4 'Sample storage' so that it now includes 'Sample storage and sample bottle size'. Lines 460-465 now read 'Another variable which differed between laboratories for the intercomparison exercise was the size of samples bottle, which ranged from 25 ml to 1 liter for the different laboratories. There was no observed difference between the methane and nitrous oxide values obtained from the various

sampling bottles and it was concluded that sampling bottles were not a controlling factor for the observed differences between laboratories. We note, however, the potential for greater air bubble contamination in smaller bottles'.

Septum We did not test for contamination (either production or adsorbtion) of methane and nitrous oxide by different septa. There are at least two recent articles presenting evidence that storing trace gas samples in bottles with rubber septa can cause contamination for methane (Magen et al., 2015, Niemann et al., 2015). The article by Magen et al (2014) also highlights the possibility of cleaning the septa, although they did not see any difference when this was conducted (albeit over an eight day period). We have amended the manuscript to address the issue of potential septa-derived contamination. This is included in the Discussion in Section 4.3 under General Recommendations. Lines 589-598 now read 'This study also revealed that sample storage time can be an important factor. The results from this study corroborate the findings of Magen et al. (2014) who showed that samples with low concentrations of methane are more susceptible to increased values as a result of contamination. The contamination was most likely due to the release of methane and other hydrocarbons from the septa (Niemann et al., 2015). Since the release of hydrocarbons occurs over a period of time, it is recommended to keep storage time to a minimum and to store samples in the dark. It should be noted that sample integrity can also be compromised due to other factors including inadequate preservation, outgassing, and adsorption of gases onto septa. For all of these reasons, it is recommended to conduct an evaluation of sample storage time for the environment that is being sampled.'

Magen, C., Lapham, L. L., Pohlman, J. W., Marshall, K., Bosman, S., Casso, M., and Chanton, J. P.: A simple headspace equilibration method for measuring dissolved methane, Limnol. Oceanogr.: Methods, 12, 637–650, 2014.

Niemann et al. (2015) Toxic effects of lab-grade butyl rubber stoppers on aerobic methane oxidation Limnol. Oceanogr.: Methods 13, 2015, 40–52

Storage time We have improved the wording of this section and Lines 448-459 now read 'Because prolonged storage of samples can influence dissolved gas concentrations, including methane and nitrous oxide, the intercomparison dataset was analyzed for sample storage effects (Table S5 in the Supplement). It should, however, be noted that assessing the effect of storage time on sample integrity was not a formal goal of the intercomparison exercise and replicate samples were not analyzed at repeated intervals by independent laboratories, as would normally be required for a thorough analysis. Nonetheless our results did provide some insights into potential storage-related problems. Most notably, there were indications that an increase in storage time caused increased concentrations and increased variability for methane samples with low concentrations, i.e. PAC1 and PAC2 samples which had median methane concentrations of 0.9 and 2.3 nmol kg-1, respectively (Fig. 7). In comparison, for samples of nitrous oxide with low concentrations there was no trend of increasing values as observed for samples with low methane concentrations.'

Page 7, line 171-173: Was there a difference between sampling bottles? No difference between sampling bottles was observed. This is now noted in the document on Lines 323-327 'Analysis conducted by the University of Hawaii of methane and nitrous oxide from each Niskin-like bottle used in the Pacific Ocean sampling did not reveal any bottle-to-bottle differences. Furthermore, analysis by Newcastle University showed there was no difference between the first and the last set of samples collected from the 1000 L tank used in the Baltic Sea sampling.'

Page 7, line 178: Which kind of stopper? Also, what is the effect of different stoppers/seals used during storage? Are some stoppers/seals leaking more than others? These questions are answered separately below

Which kind of stopper? The 1 l glass bottles used a ground-glass stopper and Apiezon grease as widely used for dissolved inorganic carbon samples.

Also, what is the effect of different stoppers/seals used during storage? Are some

stoppers/seals leaking more than others? The recent publication by Niemann et al (2015) reported on the release of organic contaminants of five different commercially available, lab-grade butyl stoppers. Different stoppers release varying quantities of different compounds. It should be noted that the objective of the Niemann et al. (2015) study was to look at the effect on biological rate measurements (methane oxidation) and not concentrations. Magen et al (2014) also looked at the potential contamination by two stoppers, although their incubation period was for 3 days only.

Page 7, lines 180-182: They used mercuric chloride for preservation, which is probably acceptable for water-column samples. However, mercuric chloride is toxic and difficult to ship and use at sea due to safety concerns. Future efforts should test alternative types of preservatives (sodium hydroxide, formaldehyde) to evaluate their suitability to preserve these samples in different marine environments. Also, mercuric chloride might not be suitable for some marine samples as Ostrom et al (2016) suggest that it could enhance nitrous oxide production by chemodenitrification in Fe-rich environments.

The reviewer raises the point that there are alternative preservatives to mercury(II) chloride. The issue with any preservative is to balance effectiveness at ceasing all relevant microbial activity, while minimizing toxicity from a human health and environmental perspective. In recent years, there have been a series of papers (Magen et al., 2014, Bussmann et al., 2015, Gloël et al., 2015) which have tested some of the alternatives to mercury(II) chloride. These include sodium azide, sodium, hydroxide, sulfuric acid, potassium hydroxide, benzalkonium chloride, and zinc chloride. These studies demonstrate the potential for alternative preservatives and show their effectiveness for a particular environment over a particular timeframe. However, they do not prove the applicability over a broad range of conditions, microbial communities, and storage times. The studies also do not provide a recommendation for the most superior preservative, nor do they always test both methane and nitrous oxide, and other substances such as dissolved inorganic carbon. Therefore, while we agree that

alternatives exist, they have not been extensively proven to be superior to the well-established use of mercuric chloride. After talking to a number of scientists about this issue, we understand that the community of scientists focused on dissolved inorganic carbon measurements are looking very carefully at alternatives to mercury(II) chloride. We have requested that measurements of methane and nitrous oxide be included in planned future tests of alternative preservatives. This will allow the whole community to switch to alternative preservatives at the same time. We have revised the manuscript to reflect our perspectives and Lines 188-193 now read 'The choice of mercuric chloride as the preservative for dissolved methane and nitrous oxide was based on its long history of usage. It is recognized that other preservatives have been proposed (e.g. Magen et al., 2014, Bussmann et al., 2015), however pending a community-wide evaluation of their effectiveness over a range of microbial assemblages and environmental conditions for both methane and nitrous oxide, we recommend continuing with a long-established method.' Magen, C., Lapham, L. L., Pohlman, J. W., Marshall, K., Bosman, S., Casso, M., and Chanton, J. P.: A simple headspace equilibration method for measuring dissolved methane, Limnol. Oceanogr.: Methods, 12, 637–650, 2014. Bussmann, I., Matousu, A., Osudar, R. and Mau, S., 2015. Assessment of the radio 3H‐CH4 tracer technique to measure aerobic methane oxidation in the water column. Limnology and Oceanography: Methods, 13(6), pp.312-327. Gloël, J., Robinson, C., Tilstone, G.H., Tarran, G. and Kaiser, J., 2015. Could benzalkonium chloride be a suitable alternative to mercuric chloride for preservation of seawater samples?. Ocean Science Discussions, 12(4), pp.1953-1969. Page 8, line 188: I assume this tank was gas tight? The tank was sufficiently gas-tight for our purposes. The tank was made of high density polyethylene (same material as used for very large carboys). Prior to sampling, the seawater was gently stirred to ensure homogeneity. Subsampling was conducted from a port located at the lowest part of the tank and approximately one-tenth of the tank's contents were sampled. A headspace was created during the sampling and by the time the last sample was collected, there was approximately a 1 meter distance between the sampling port and the headspace interface.
Page 8, lines 196-198: Was there a difference between this first and last samples? Any change in temperature during sampling would affect gas concentrations. Also, I suppose a headspace was created in the 1000 L water tank as samples were drawn? No difference was observed between the first and last samples. Please see our description about sampling from the tank in our previous response.

Page 9, lines 223-225: "headspace collected into a gas tight syringe and injected": How is this different than the physical injection? This sentence highlighted the fact that the headspace had been subsampled into a separate syringe. However, this is a very subtle point and as the Reviewer points out, by including physical injection in the previous sentence, this extra description is not needed. We have removed this sentence from the manuscript.

Page 9, lines 228-229: How many standards were typically used? The number of standards used by each laboratory ranged from 2-4. This information is provided in the Supplementary Information in Tables 6 and 7.

Page 9, line 248: Why does the tubing need to be maintained at low temperatures? The majority of scientists install gas purifiers on the gas supply lines which feed any gas analyzer. This is a preventative measure in case the commercially sourced compressed gas cylinders vary in quality, which can occur for even the high-purity gases. The majority of the gas purifiers are commercially available, however a homemade purifier consisting of a length of tubing packed with Porapak or Hayesep material and immersed in liquid nitrogen is recommended for methane analysis when measurements are made using purge-and-trap. The larger volume of purge gas used during purge-and-trap causes trace contaminants to become concentrated which affects the methane chromatogram. This does not appear to be an issue when analyzing methane using the headspace equilibrium technique. We have improved the text to clarify these additional steps for methane analysis. Lines 257-260 now read 'In addition to commercially available scrubbers, purification of the sparge gas was achieved by passing it through stainless steel tubing packed with Poropak Q and immersed in liquid nitrogen. This is a recommended precaution to consistently achieve a low blank signal of methane.'

Page 9, line 249: Low blank for what? Methane, nitrous oxide, or both? We have clarified this in response to the previous comment.

Page 10, line 251-252: Be more specific: "liquid nitrogen (-165oC) for methane or cooled ethanol (-70oC) for nitrous oxide." This sentence has been improved and lines 262-263 now read 'Cryotrapping was achieved for methane using liquid nitrogen (-195oC) and either liquid nitrogen or cooled ethanol (-70oC) for nitrous oxide.'

Page 11, line 303: By "comparable values" do you mean peak area? Not quite. The text has been improved to make this clearer. Lines 314-315 now read 'For the two laboratories with an in-house standard of comparable mole fraction to the WRS, an offset of 3% and a >20% offset was reported.'

Page 13, lines 362-371: This point comes across more clearly in the Fig. 3's legend. Perhaps rewrite? We agree this section was awkwardly written and Lines 376-382 now read 'The relevance to final methane concentrations is demonstrated by considering the values reported by the University of Hawaii for PAC2 samples (Fig. 1b). An almost 30% increase in final methane concentration occurs from the use of the calibration equation in Figure 3c, compared to Figure 3a. This derives from a measured peak area for methane of 62 for a sample with a volume of 0.076 L and a seawater density of 1024 kg m-3, yielding a final methane concentration of 2.1 and 2.8 nmol kg-1 using the equations from Figure 3a and 3c, respectively.'

Page 14, lines 388-401: A sample with higher nitrous oxide concentrations could also be used in future intercomparison efforts. For instance, nitrous oxide concentrations of up to 1000 nmol/L were measured in coastal waters off Peru (Arévalo-MartÍnez et al.,2013). The intercomparison of methane and nitrous oxide used typical shipboard sampling procedures as it replicated typical sampling and storage procedures. Future intercomparison exercises will have the ability to manipulate concentrations of methane

and nitrous concentrations. The University of Hawaii is awaiting delivery of a large (200 liter) equilibrator unit. The 200 L capacity is smaller than the 760 L equilibrator used to produce reference material for dissolved inorganic carbon by Andrew Dickson, but it will allow us to produce reference material of varying concentrations on demand.

Page 15: Why was the variability higher for the BAL5 dataset? Could this be related to sampling and/or storage? The BAL5 samples had the highest concentrations of nitrous oxide sampled from the Baltic Sea and were associated with high inter-laboratory variability. We believe that the high variability is caused to a large extent by the non-linear response of the ECD. Differences in calibration procedures by the different laboratories, as shown in Figure 6, become exacerbated for high concentrations of nitrous oxide. If sampling and/or storage were the primary causes of the variability, we would have expected to see equally high variability in the samples with lower concentrations.

Page 16, lines 438-439: Was this only true for samples with methane concentrations less than atmospheric concentrations? Yes, it appears that low concentration samples are more susceptible to an increase due to contamination.

Page 18, line 512: What would be their maximum recommended storage time? For samples with very low or high concentrations, analysis within 2 months is recommended. For samples with concentrations equivalent to or exceeding atmospheric equilibrium, analysis could be conducted within a slightly longer timeframe e.g. 6 months.

Page 19, lines 532-534: They discuss detection limits for methane but not for nitrous oxide analysis methods. What are the detection limits associated with the two different analysis methods (headspace equilibration versus purge and trap)? We report on lines 550-552 that 'The low concentrations of nitrous oxide still exceed detection limits by at least an order of magnitude for even the less-sensitive headspace method due to the high sensitivity of the ECD.' In response to an earlier comment by Reviewer 1, we have now included a brief comparison of the detection limits for headspace equilibrium and purge-and-trap in the Introduction and Lines 89-95 now read 'The purge and trap technique is typically more sensitive by 2-3 orders of magnitude over headspace equilibrium. However, the purge and trap technique requires more time for sample analysis and it is more difficult to automate the injection of samples into the gas analyzer. Headspace equilibrium sampling is most suited for volatile compounds that can be efficiently partitioned into the headspace gas volume from the seawater sample. Its limited sensitivity can be compensated by large volume analysis e.g. (Upstill-Goddard et al., 1996).'

Page 20, lines 560-565: Other important points, e.g., sample volume, septa/seals used, preservative used, should also be included in future efforts. We agree with this comment, and have modified Section 4.3 in the Discussion to address this point.

Page 20, line 576-577: This assumes that the air in the laboratory where the measurements are done is not contaminated by other sources of nitrous oxide (non-atmospheric). We agree with this comment which is why we also suggested using air from compressed gas cylinder after cross-checking its concentration. This is more likely to be relevant for methane than nitrous oxide. Lines 605-607 read 'The air used in the equilibration process could be sourced from the ambient environment if sufficiently stable or from a compressed gas cylinder after cross-checking the concentration with the appropriate gas standard.'

Page 20, line 586: Bourbonnais et al. (2017) also used air-equilibrated seawater standards to calculate water-column nitrous oxide concentrations off Peru. Thank you for this reference, it is now included in the manuscript

Figures 1: Are values of methane at atmospheric equilibrium expected at 25 m depth? Is this in the mixed layer? At Station ALOHA, the mixed layer depth nearly always exceeds 25 m during the winter months (November-March). During the expedition in February 2017 when the samples were collected, the mixed layer depth ranged from 110-130 m. We have now reported this in the text on Lines 176-177.

Figure 7: Are these relationships significant (add r2)? Ideally, to assess storage effects, samples collected the same way and using the same analysis method should be analyzed at different time points by the same laboratory. The r2 value is included for each of the regression lines shown in Figure 7a and 7b. We completely agree with the Reviewer's comment that the same laboratory needs to conduct a time-course set of measurements for a thorough analysis of storage effects. This was not part of the intercomparison work, but is clearly needed for a Best Practice Guide which is being planned.

Tables 6 and 7: Add detection limits for each laboratory. We considered including detection limits, but did not include them in this Supplementary Table. This is because detection limits can be lowered (improved) by increasing the sample volume (for purge-and-trap method) or altering the ratio of water to headspace (for the headspace equilibrium method). In Column 3 of Tables 6 and 7, published references have been included for the majority of the laboratories. These include more in-depth description of the individual methods than can be provided here.

Add last name "Macarena Burgos" as done for all other researchers. Done

Page 4, lines 76 to 78: Typically is used twice in these two sentences – remove one instance. Changed

Page 18, line 501: change "equilibration" for "equilibrated". Changed

Page 19, line 545: change to "switching between different calibration curves." Changed

---

## Author Comment (AC2) · 31 Aug 2018

Referee #2 In their manuscript, Wilson et al. present data from a recent international intercomparison study which evaluated the analytical procedures used to measure the concentrations of methane and nitrous oxide dissolved in seawater. Specifically, seawater samples and gaseous standards were sent to several different laboratories for analysis. Since the measurement of methane and nitrous oxide concentrations are mainly done in the gas, not liquid, phase, the different laboratories had different protocols to first separate the dissolved gas prior to analysis as well as the final analysis; while the different labs had different protocols, they mainly involved either headspace equilibration or a purge and trap technique. The results of this intercomparison are striking, with different laboratories reporting concentrations that could be different by

several hundred percent. The highest percent differences were reported for the lowest concentration samples, and since low concentrations are typically reported in the near-surface waters, this inter-laboratory difference is particularly troubling for global extrapolation of sea-to-air fluxes for these two gases. The impact of this manuscript is that it identifies significant inconsistencies between laboratories, and while the data from any one laboratory is likely valid for testing hypotheses, combining data from multiple laboratories for global extrapolation or time series analysis will lead to significant unknowns. A the end of the manuscript, the reader is left hungry for more, wondering how these inconsistencies might be rectified with a hypothetical Standard Operating Procedure. But while the authors provide a few recommendations for how to lower uncertainties, they do not prove the major cause of these inconsistencies, and thus which procedure might be preferred. The authors appropriately did not attempt this recommendation as it was beyond what their data can illuminate. For example, a full analysis of the headspace equilibration procedure would require each laboratory to establish the accuracy and precision of each variable in Equation 1 (pressure, temperature, salinity, headspace volume, and water volume) using their procedures. The authors assess the calibration of the analytical instrument and the variability of the overall results, but not these specific variables. In addition, the authors recognize that storage time is a variable significantly influencing the results. Since these additional variables were not systematically investigated, the authors are correct in not recommending a preferred procedure, and instead choose to report overall inconsistencies. We thank Reviewer #2 for their comments. We are building on the results from this intercomparison exercise and in the future will have a Best Practice Guide for the measurements of dissolved methane and nitrous oxide.

Sample storage: I recommend that the authors expand section 3.4. I found this section too brief on experimental details and I was left assuming how storage time was assessed. Was the sample storage time variable controlled in any systemic way or is this simply the time it took different labs to actually conduct their analyses? Is there any way to normalize the data in Figures 1 and 4 to sample storage time or would that be extending this data too far? Can the authors assess how much variation in the dissolved concentrations is due to storage vs. procedure? The specific questions are answered separately below. In response to the general comment, we have re-structured Section 3.4 to improve its clarity. Lines 448-459 now read "Because prolonged samples storage can influence dissolved gas concentrations, including methane and nitrous oxide, the intercomparison dataset was analyzed for sample storage effects (Table S5 in the Supplement). It should, however, be noted that assessing the effect of storage time on sample integrity was not a formal goal of the intercomparison exercise and replicate samples were not analyzed at repeated intervals by independent laboratories, as would normally be required for a thorough analysis. Nonetheless our results did provide some insights into potential storage-related problems. Most notably, there were indications that an increase in storage time caused increased concentrations and increased variability for methane samples with low concentrations, i.e. PAC1 and PAC2 samples which had median methane concentrations of 0.9 and 2.3 nmol kg-1, respectively (Fig. 7). In comparison, for samples of nitrous oxide with low concentrations there was no trend of increasing values as observed for samples with low methane concentrations.'

Was the sample storage time variable controlled in any systemic way or is this simply the time it took different labs to actually conduct their analyses? The sample storage time represents the time taken for different laboratories to conduct the analysis. There was no control of the storage time.

Is there any way to normalize the data in Figures 1 and 4 to sample storage time or would that be extending this data too far? We would be uncomfortable doing this conversion because it would insinuate a higher influence of sample storage on concentrations than what we can currently prove. We refer the readers to Figure 7 which shows concentration and coefficient variation against storage time for the samples with the lowest concentration of methane.

Can the authors assess how much variation in the dissolved concentrations is due to storage vs. procedure? This would require a time-course set of measurements

which was not conducted as part of this exercise. This would be a very interesting experiment and could feature in future intercomparisons. What we have noted in our response to the overall comment, is that contamination is considered most likely for the samples of methane collected from the Pacific Ocean. These samples had methane concentrations of 0.9 and 2.3 nmol kg-1 and therefore were most sensitive to release of small quantities of hydrocarbons by the septa.

The authors suggest that leakage may be a source of uncertainty for longer storage times, but they don't raise the possibility of inadequate preservation. Most groups analyzing these dissolved gases assume that adding enough mercuric chloride to a sample will halt all biological activity, but that may not be the case. In addition, what is the chance that gases are outgassing or adsorbing to the stopper? Since these are both possible influences on the final results, I suggest that the authors also briefly raise these possibilities. In response to the comments made by Reviewer #2, we have restructured the relevant part of the Discussion to specifically address the issue of sample storage. Lines 589-598 now read 'This study also revealed that sample storage time can be an important factor. Specially, the results from this study corroborate the findings of Magen et al. (2014) who showed that samples with low concentrations of methane and more susceptible to increased values as a result of contamination. The contamination was most likely due to the release of methane and other hydrocarbons from the septa which interfere with the dissolved methane in the sample (Niemann et al., 2015). Since the release of hydrocarbons occurs over a period time, it is recommended to keep storage time to a minimum and to store samples in the dark. It should be noted that sample integrity can also be compromised due to other factors including inadequate preservation, outgassing, and adsorption of gases onto septa. Due to all of these reasons, it is recommended to conduct an evaluation of sample storage time for the environment that is being sampled.'

Please note that in response to comments by Reviewer #1 we addressed the issue about alternatives to mercuric chloride and Lines 188-193 now read 'The choice of mercuric chloride as the preservative for dissolved methane and nitrous oxide was due to its long history of usage. It is recognized that other preservatives have been proposed (e.g. Magen et al., 2014, Bussmann et al., 2015), however pending a community-wide evaluation of their effectiveness over a range of microbial assemblages and environmental conditions for both methane and nitrous oxide, we recommend continuing with a long-established method.'

Overall, this investigation appears robust and the manuscript is well written. The authors have uncovered a significant result which will benefit the community. Thank you for your comments

---

## Author Comment (AC3) · 31 Aug 2018

Reviewer #3 The authors present a very important result of an intercomparison between many labs for measuring methane and nitrous oxide levels in ocean water samples. Overall, I think this paper is well written and will be a great contribution to the field. A lot of planning and work went into this study, and is worthy of publishing. The main focus is to look at standards, calibration issues, but don't really address how with the large variability of how people process water samples affects the results. I think this paper highlights some very important issues regarding trace gas analysis in open ocean settings, and could be transferred to other environments. Section 4.3 will be regarded as a huge step forward, once this group is able to produce a Good Practice Guide to the community. While I was left wanting to know about how best to make

these measurements, I acknowledge that this group is on the way to doing that and will do that. This paper is the first step. The conclusion that calibration issues are a huge problem in this field, and the recommendation to produce reference material for both trace gases is a wonderful contribution.

1. They mention on line 587 for all labs to do internal checks by measuring an air-equilibrated seawater. They mention needing a water bath and stirrer. Since this is a main finding that could be implemented in the community ASAP, could they provide true details of the setup? This might be appropriate in the supplementary materials. We reference four studies which report using air-equilibrated seawater as an internal control. Each of these studies had slightly different procedures and at this stage we refer the readers to these publications for further information. We would like to conduct a more thorough analysis of how robust these measurements are (e.g. sensitivity to temperature fluctuations) before publishing more detailed recommendations as part of a planned Best Practice Guide.

2. Line 220: Why is there such variation in equilibration time for the gases; between 20 min to 24 hours? Has anyone done a time series of equilibration times to show what the time needs to be? This could be part of the recommendations. The longer equilibration times are due to overnight equilibrations in water baths. All laboratories should test equilibration time for the headspace analysis or the sparge time for the purge-and-trap technique, when establishing their own personal protocols for different sample volumes, temperatures, and sampling habitat.

3. Line 272: Where do the CV values come from that are plotted in figure 7b? In table S2, there is one column for "mean CV" which seems to be related to each lab, and not specifically for PAC1 and PAC2. Maybe those CVs are just not reported in the table, in which case, please report them. The values of coefficient of variation (%) shown in Figure 7b are associated with methane concentrations measured by each lab for PAC1 and PAC2 samples (collected in February 2017). These specific values are not included in any of the Supplementary Material tables, where we instead report

the mean coefficient of variation associated with each laboratory. We also report the coefficient of variation for the whole batch of samples in Table 2 in the main document.

4. Line 371: it is not clear to me what they mean by "sample contamination, discussed below (datasets J and K)." Where do they discuss below? Could they call out the specific sections they want the reader to refer to? This sentence has been improved and Lines 385-388 now read 'In contrast, the datasets with a higher offset at low methane concentrations (Datasets J and K) could be due to the use of incorrect intercepts as well as other factors including sample contamination, discussed in Section 3.4.'

5. Line 430 and on: The storage section really added a nice dimension to the paper, even though it was not a main focus. On line 445, you state that BAL2 shows a decrease in N2O concentrations over time. Can you show that graph? When graphed, I see that BAL2 shows an increase with time but it also seems within the variability of the measurements. Reviewer #3 has highlighted an error in the manuscript as we meant to say BAL5, not BAL2. We apologize for the error. Because there is not a significant decrease of nitrous oxide with time, we did not initially include this Figure in the manuscript. We now feel that it is inappropriate to include this comment and we have removed the sentence 'There was some indication of a decrease in concentration for seawater samples with higher concentration of nitrous oxide (i.e. BAL5), which could have been caused by gas leakage' from the manuscript.

6. Line 432: The explanation of the results from Magen 2014 are a bit misleading. That paper shows that at methane concentrations less than ~1ppm in the headspace, there could be a storage issue after 1 year. And the issue is that concentrations increase. There should be more context to your statement "because prolonged sample storage adversely affects dissolved methane and nitrous oxide samples (Magen et al., 2014)...." In response to this comment and comments from other Reviewers, this section has been rewritten and Lines 448-459 now read 'Because prolonged samples storage can have an adverse affect on dissolved gases, including methane and nitrous oxide, the intercomparison dataset was analyzed for sample storage effects (Table S5

in the Supplement). It should however be noted that assessing the effect of storage time on sample integrity was not a formal goal of the intercomparison exercise and replicate samples were not analyzed at repeated intervals by independent laboratories, as would normally be required for a thorough analysis. Nonetheless our results did provide some insights. Most notably, there were indications that an increase in storage time caused increased concentrations and increased variability for methane samples with low concentrations, i.e. PAC1 and PAC2 samples which had median methane concentrations of 0.9 and 2.3 nmol kg-1, respectively (Fig. 7). In comparison, for samples of nitrous oxide with low concentrations there was no trend of increasing values as observed for samples with low methane concentrations.'

7. Line 439: Storage for methane. Where did the data come from for figure 7? From the supplemental tables, the only storage time data shown is from Feb for PAC 2, and Nov for PAC1. Just from a first look, there are only 7 reported values for methane for PAC1 Nov in Table S2, but 11 points plotted in figure

The questions in 7, 7a, and 7b are dealt with below

7a. Where is the extra data coming from? Data looks consistent for PAC2. If I replot the storage days from table S5 vs the concentrations from table S2, I get the following graphs. (For the graphs below, methane concentrations are plotted over storage time with the outliers and without.) Those outliers were identified in figure 7a with () around the symbols, which is stated in the figure cation to be taking out of the regression. For PAC2, I reproduce what was reported in figure 7a, but for PAC1, the story is completely different. Please address this inconsistency.

a. After agonizing over the mismatch of this data, it looks like they plotted PAC1 Feb 2017 in figure 7a, not PAC1 Nov 2013. If that's the case, the storage time data presented in table S5 is not right.

I think the confusion exists because the Supplementary Table 5 included the storage times for samples collected in November 2013 (Pacific_1) and February 2017 (Pacific_2). However, we also referred to the sampling depths as PAC1 (25 m depth) and PAC2 (700 m depth). Therefore, there is too much similarity between date (Pacific_1 and Pacific_2) and depth (PAC1 and PAC2). After consideration, we have removed the column in Table S5 which lists the storage time for the November 2013 samples. Since we do not refer to the November 2013 samples in the main document, there is no loss of information by not including their storage times and there will be less confusion.

The data used to create Figure 7 is included in the attached pdf

Where did the data come from for figure 7? None of the November 2013 Pacific_1 data are shown in Figure 7. We state on Lines 167-171 that 'The November 2013 samples are included in Figure S1 and S2 in the Supplement, but are not discussed in the main Results or Discussion because fewer laboratories were involved in the initial intercomparison, and the results from these samples support the same conclusions obtained with the more recent sample collections.' To make this clearer for the readers, we have repeated this text in the Figure 7 legend and Line 909 now reads '….collected in February 2017'

Where is the extra data coming from? There are no extra data. For the February 2017 Pacific_2 Column in Table S5 there are 14 labs in total and 2 of these labs (Red and Beige) did not measure methane in the Pacific Ocean. The 12 datasets are represented by the 12 data points are shown in Figure 7.

8. Can you add a column in the supplemental table for N2O for how each person dealt with water, like what was done for methane? Water is a huge issue for N2O precision, and there is no mention of how water was dealt with. This is now included in Supplementary Table 7. As a quick response, water vapor is removed by most laboratories using a drying agent frequently in combination with Nafion tubing.

9. Line 507, if your intent is to show some examples, you should add "for example" to your reference list here. There are many other papers that show this. Changed

10. Line 557: extra space between "proposed" and "production" Changed

11. In table S5, "red" is listed as having measured something on the PAC samples 140 days after collection. But when I try to cross reference this in table 2, it looks like "red" didn't measure for methane. It might help to know if the storage times in table S5 are for methane and/or N2O. Overall, I think this table needed revisiting. Reviewer#3 is correct, 'red' Laboratory M only made nitrous oxide measurements. There was also one laboratory (Laboratory D, beige) that only measured methane. We have improved the Table heading to make this clearer and it now reads 'The reported storage times are for both methane and nitrous oxide (Laboratory M 'red' measured methane only and Laboratory D 'beige' measured nitrous oxide only).'

12. Figure S1, what is the gray dashed line? What do colors represent? Individual data points are plotted sequentially in increasing value with the same color symbol for each laboratory in all plots for the main text and Supplementary Material. The dashed grey line represents the value of methane at atmospheric equilibrium as stated in the Figure legend.

13. Figure S2, are a and b shallow water and c and d deep water? Make that clear in the first description of the figure. It says "same location" but what you mean is at the same lat/long but two different depths. Also, caption says "In contrast, the concentration of nitrous oxide in the deep-water samples (Figure S2c and d) was more consistent and the data values for the laboratories that measured samples from 2013 and 2017 are shown together in Figure S2d." is that also supposed to be shown by a gray dashed line? Can you make the scales the same for both sides? We have now plotted Figure S2c on the same scale as Figure S2d. Each subplot also includes a description of depth as well as the actual Figure legend. The Figure S2 legend has been improved and now reads 'Supplementary Figure S2: Nitrous oxide concentrations in seawater samples collected at the same location but varying depths in the North Pacific Ocean on February 2017 (Fig. S2a and c) and November 2013 (Fig. S2b and d). The dashed grey line represents the value of nitrous oxide at atmospheric equilibrium for the 25 m

seawater samples (Figure S2a and b). The February 2017 plots are discussed in the main manuscript and are replicated here to facilitate comparison with the November 2013 data, particularly for comparison with the 700 m samples (Figure S2d).'

14. Supp table 1: what is the point of the far right columns in this table? What is the mean CV of? For example, for lab A, it says 9.2% CV. Did you take CV for each BAL1, BAL2, etc, and then average that? Since we don't see the BAL1 CV, this is not clear. That being said, I'd like to see the CV for the standards run in the lab. From my experience with N2O, I can have ∼10% CV if there is still water in the sample. The purpose of the Supplementary Tables 1-4 is to provide further information about the data values provided in Figure 1 and Figure 4 in the main document. The far right-hand columns provide a measure of variability for each laboratory as shown by the mean coefficient of variation (%) and the mean offset (%). We now state in the Table heading that these values are for all sampling stations shown in each respective Table, 'based on all 7 sampling stations'. Reviewer #3 also indicates that it would be helpful to see the coefficient of variation (%) for standards as well as the samples. In our experience, there is always higher precision associated with analysis of standards. This is because sample analysis includes multiple steps of sample handling, gas extraction/equilibration. Therefore we prefer to report the precision associated with sample analysis, as the precision associated with standards will be lower than this value.

Please also note the supplement to this comment:
https://www.biogeosciences-discuss.net/bg-2018-274/bg-2018-274-AC3-supplement.pdf